# Orexin population activity precisely reflects net body movement across behavioral and metabolic states

Alexander L Tesmer[1]*, Paulius Viskaitis[1], Dane Donegan[1], Eva F Bracey[1], Nikola Grujic[1], Tommaso Patriarchi[2,3], Daria Peleg-Raibstein[1,3], Denis Burdakov[1,3]*

[1]Swiss Federal Institute of Technology (ETH Zürich), Department of Health Sciences and Technology, Zürich, Switzerland; [2]Institute of Pharmacology and Toxicology, University of Zürich, Zurich, Switzerland; [3]Neuroscience Center Zürich (ZNZ), Zürich, Switzerland

## eLife Assessment

This **important** study shows that the activity of hypothalamic hypocretin/orexin neurons (HONs) correlates with body movement over multiple behaviors. **Compelling** evidence, supported by sophisticated, cutting-edge tools and data analyses, highlights a link that appears to be unique to HONs. This work should be of interest to scientists studying peptidergic neurons, movement, energy regulation, and brain-body coordination.

*For correspondence:
alexander.tesmer@hest.ethz.ch
(ALT);
denis.burdakov@hest.ethz.ch
(DB)

Competing interest: The authors declare that no competing interests exist.

**Abstract** Tracking net body movement in real time may enable the brain to estimate ongoing demands and thus better orchestrate muscle tone, energy balance, and arousal. To identify neural populations specializing in tracking net body movement, here, we compared self-initiated movement-related activity across genetically-defined subcortical neurons in the mouse brain, including dopaminergic, glutamatergic, noradrenergic, and key peptidergic neurons. We show that hypothalamic orexin/hypocretin-producing neurons (HONs) are exceptionally precise movement-trackers, encoding net body movement across multiple classified behaviors with a high degree of precision, independent of head acceleration. This tracking was so precise that video analysis of the mouse body movement reliably served as a low-cost biometric for HON population activity. The movement tracking was independent of internal nutritional states, and occurred in a communication bandwidth distinct from HON encoding of blood glucose. At key projection targets, orexin/hypocretin peptide outputs correlated with self-initiated movement in a projection-specific manner, indicating functional heterogeneity in HON outputs. Finally, we found that body movement was not encoded to the same extent in other key neural populations related to arousal or energy. These findings indicate that subcortical orchestrators of arousal and metabolism are finely tuned to encode net body movement, constituting a bridge multiplexing ongoing motor activity with internal energy resources.

## Introduction

Neural activity related to body movement recently received renewed attention enabled by modern video tracking techniques (*Stringer et al., 2019*; *Musall et al., 2019*; *Wang et al., 2023*; *Zagha et al., 2022*). Although the cortex is classically subdivided into sensory and motor regions (*Carpenter and Reddi, 2012*), movement-related signals make up a large proportion of single-neuron activity across both regions (*Stringer et al., 2019*; *Musall et al., 2019*; *Wang et al., 2023*; *Zagha et al.,*

*2022*; *Crochet and Petersen, 2006*; *Deschênes et al., 2012*; *Herrington et al., 2009*; *Keller et al., 2012*; *Martinez-Conde et al., 2000*; *Miura and Scanziani, 2022*; *Niell and Stryker, 2010*; *Salkoff et al., 2020*; *Schneider and Mooney, 2015*; *Vinck et al., 2015*; *West et al., 2022*; *Yu et al., 2016*; *Bimbard et al., 2023*). Modern technologies for recording large numbers of neurochemically-undefined neurons also detected movement-related activity across subcortical brain areas (*Stringer et al., 2019*), but suggested some differences between brain areas (*Wang et al., 2023*). This raises the question of whether population activities of neurochemically distinct subcortical neurons are shaped by the magnitude and frequency of body movements.

We explored this question by examining the population activity of hypothalamic neurons producing the peptide transmitters hypocretins/orexins (*de Lecea et al., 1998*; *Sakurai et al., 1998*). Hypocretin/orexin neurons (HONs) track the body's glycemic dynamics and hunger states (*Sakurai et al., 1998*; *Yamanaka et al., 2003*; *Burdakov et al., 2005*; *Viskaitis et al., 2024*), and orchestrate arousal (*Chemelli et al., 1999*; *Lin et al., 1999*; *Adamantidis et al., 2007*; *Sakurai, 2014*). Metrics and interpretations of HON activity are of substantial interest in both basic and clinical sciences, due to their brain-wide influence and link to diagnosis and treatment of multiple brain disorders (*Thannickal et al., 2000*; *Peyron et al., 1998*; *Thannickal et al., 2007*; *Nishino et al., 2000*; *Peyron et al., 2000*; *Johnson et al., 2010*; *Dauvilliers et al., 2020*; *Bassetti et al., 2019*; *Nishino et al., 2001*). Yet, fundamental questions remain unresolved in relation to HONs and movement:

- Does HON population activity track specific behaviors, or the general magnitude of body movement?
- Given that hunger attenuates some neural operations (*Plaçais and Preat, 2013*), does it disable HON movement tracking?
- Is movement vs glycemic information tracked by quantitatively distinct bandwidths of HON activity?
- How does HON tracking of body movement compare to other genetically defined neural subcortical clusters linked to arousal and metabolism?

Here, we address these questions by simultaneously measuring population HON activity and body movement. We report that HONs encode the instantaneous magnitude of body movement; a feature that was observed across and within unique behaviors. HON movement encoding occurs in an activity bandwidth quantitatively distinct from their glucose-tracking bandwidth and is independent of metabolic or glycemic states. We further find that orexin/hypocretin peptide release dynamics related to body movement differ between various projection targets of orexin neurons. These results define a low-cost video biometric of HONs in a widely-used experimental setting, and add to our understanding of movement-related activation of genetically distinct subcortical neural clusters.

## Results

### A general movement metric statistically explains rapid HON population dynamics across and within behaviors

To correlate HON dynamics with general body movement, we used a video camera to monitor n=15 head-fixed mice on a running wheel. Head fixation allowed us to repeatedly record movement from a fixed angle while focusing on factors other than head acceleration, as done in other recent studies (*Stringer et al., 2019*; *Musall et al., 2019*). Simultaneously, we recorded HON activity via intrahypothalamic fiber photometry of HON-selective GCaMP6s activity sensor (*Figure 1A*). Movement artifacts were controlled for via isosbestic excitation. To measure body movement, we derived a 1-dimensional movement metric by calculating the total per-pixel difference between consecutive frames, and then convolving with a decay kernel matched to the known dynamics of the GCaMP6s sensor (*Figure 1B*; *Chen et al., 2013*). We then quantified the correlation between recorded HON dynamics, movement, and running speed on the wheel. HON dynamics exhibited a positive correlation to both running speed (r=0.50 ± 0.03) and the movement metric (r=0.68 ± 0.04, *Figure 1C and D*). The movement metric's correlation was significantly stronger than running (p<0.0001). Notably, we also observed a positive correlation with the movement metric within epochs containing no locomotion on the wheel (r=0.54 ± 0.03), demonstrating that movement encoding by HON dynamics is not limited to running (*Figure 1C and D*).

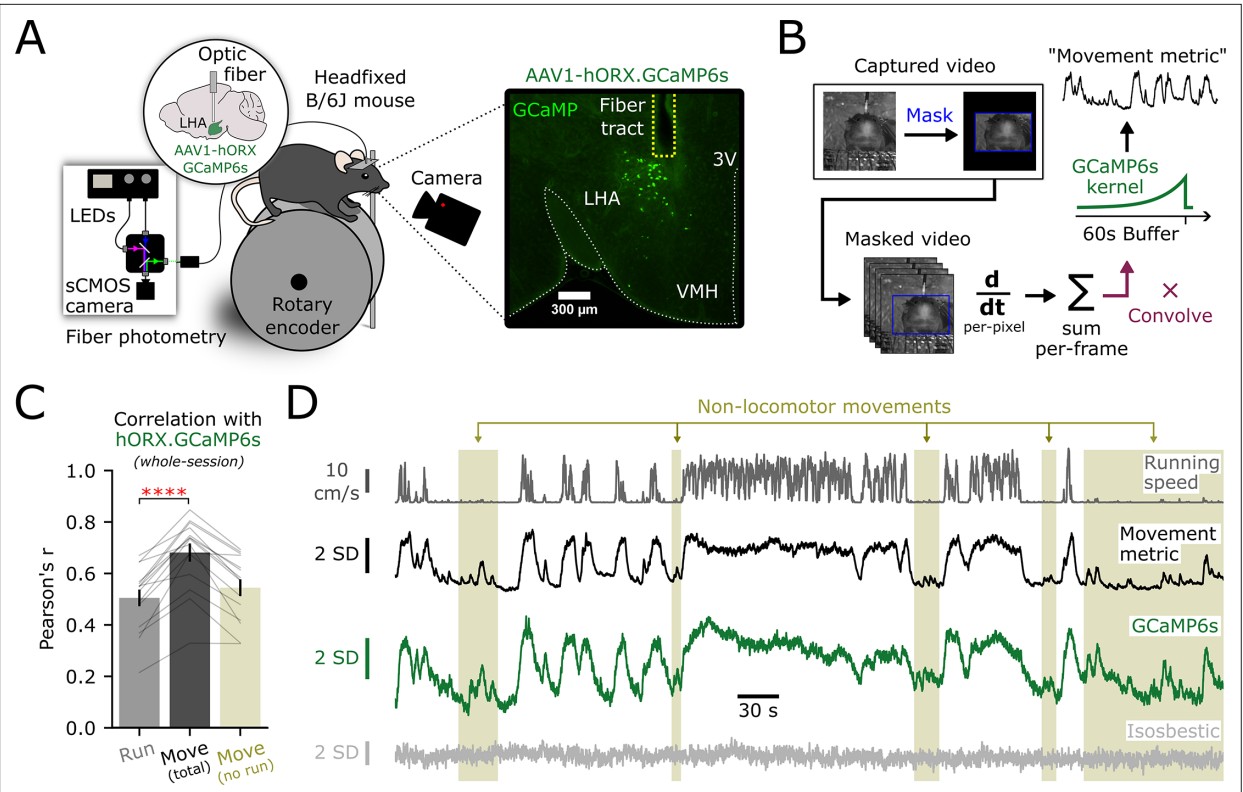

**Figure 1.** Movement magnitude is represented by rapid hypocretin/orexin neuron (HON) dynamics. (**A**) Schematic depicting the photometry system and voluntary wheel-running apparatus with video capture (left) and representative expression (right) of AAV1-hORX-GCaMP6s in HONs. LHA; lateral hypothalamic area; 3V; third ventricle; VMH; ventral medial hypothalamus. (**B**) Processing pipeline to generate a 'movement metric' from video recording of voluntary movement. (**C**) Whole-session correlation of HON photometry with simultaneously recorded behavior metrics from n=15 mice: running (r=0.50 ± 0.03), the movement metric (r=0.68 ± 0.04), the movement metric in epochs where locomotion was <1 cm·s⁻¹ (r=0.54 ± 0.03). Metrics are given as mean ± SEM, connected lines represent individual mice. Asterisks indicate comparison between running and movement metric (paired *t*-test: $t_{14}$=6.676, ****p<0.0001). (**D**) Traces from an example experiment. From top to bottom: running on the wheel, the video-derived movement metric, HON GCaMP6s photometry, and isosbestic control. Shaded beige regions indicate non-locomotor epochs where the movement metric still reported strong positive correlations with HON activity.

Does HON activity represent discrete behavioral states, or rather, encode movement magnitude generalized over behaviors? Both hypotheses could potentially explain the correlation in *Figure 1*. To compare our movement metric (*Figure 1B*) across different behaviors, we trained a deep-learning network (see methods) to classify video recordings into the five most commonly observed behaviors: resting, running, grooming, chewing, and sniffing (*Figure 2A*). Recorded HON activity was the highest during running, and lowest during resting showing significantly different activity levels across behaviors (p<0.0001, *Figure 2B and C*). The average HON activity during specific behaviors followed a nearly 1:1 relationship against its average movement metric (*Figure 2D*). We then asked whether HON activity could be better explained by a classified behavior, or if it simply followed the movement metric regardless of the specific behavior. To test this, we fit three models in which HON activity was predicted by (1) the classified behavior, (2) the movement metric, or (3) the movement metric as an interaction effect with classified behavior (*Figure 2E*). Using the Akaike information criterion to assess model fit, we found model 1 was vastly inferior to model 3. However, the strength of this improvement was substantially less when comparing model 2 to model 3. These results suggest that HON activity is tracking the magnitude of movement change, not encoding different types of behavior. Finally, we sought to analyze HON dynamics during transitions between behavioral states. We first identified the six most common behavioral transitions (*Figure 2F*) and aligned them to both the movement metric and HON activity. As expected, the HON activity strongly resembled the magnitude of body movement exhibited along behavioral transitions (*Figure 2G and H*).

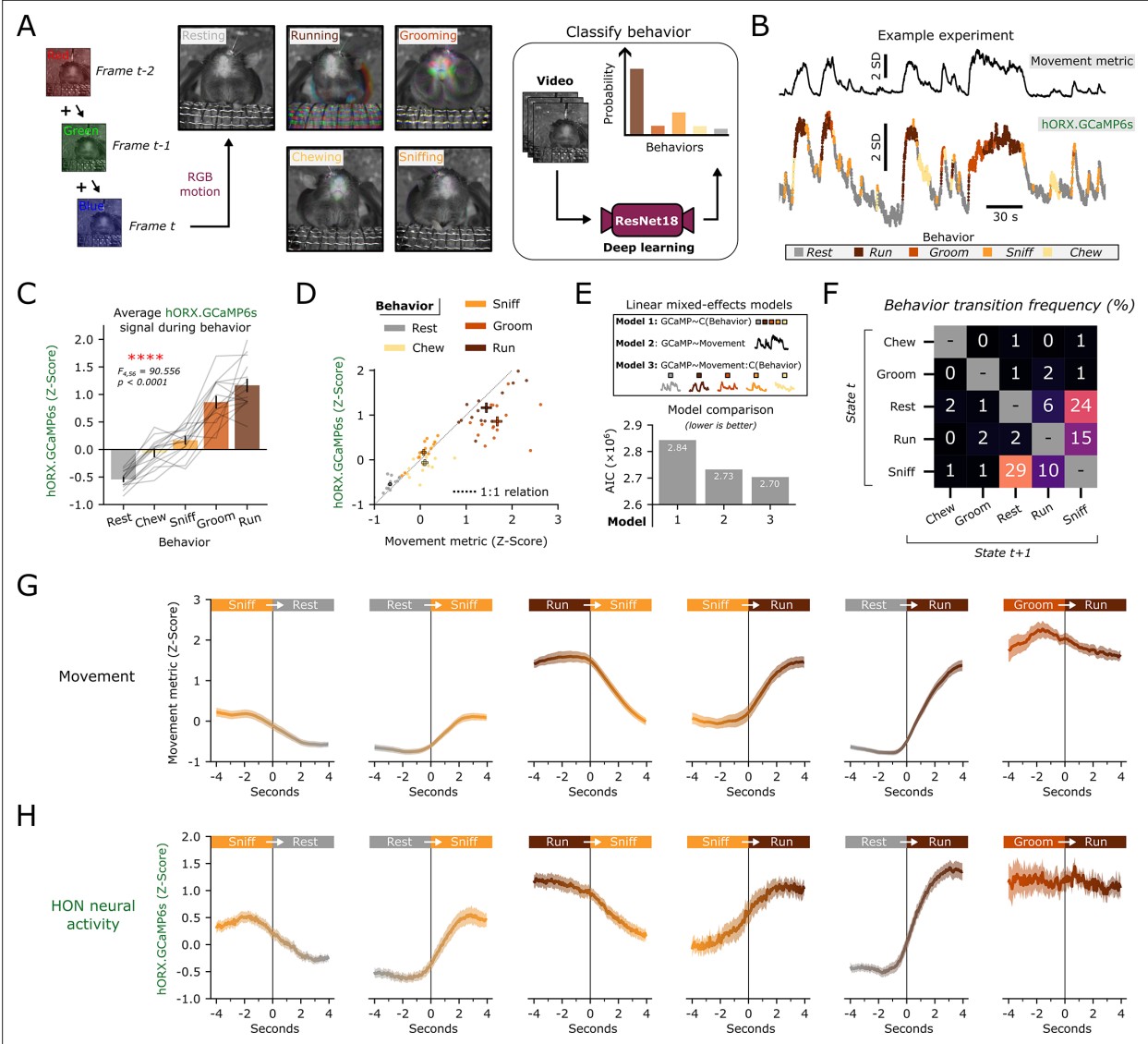

**Figure 2.** Hypocretin/orexin neuron (HON) dynamics represent movement magnitude across classified behaviors. (**A**) Diagram for classifying behaviors from video recordings via deep learning. (**B**) Example experiments showing movement (upper, black) and color-coded photometry (lower) after behavioral classification. (**C**) Average normalized HON GCaMP6s signal across classified behaviors in n=15 mice (rmANOVA: $F_{4,56}$=90.556, ****p<0.0001). (**D**) Average value of the normalized HON GCaMP6s signal from n=15 mice plotted against the average value of the movement metric in each classified behavior. Thick bars represent SEM across both metrics. The dotted line represents an idealized 1:1 linear relationship between movement and photometry with an intercept at zero. (**E**) Comparison of three linear mixed effects models using movement and/or behavioral classification to predict HON GCaMP6s signal, quantified by Akaike information criterion. (**F**) Average behavioral transition matrix for n=15 mice. (**G**) Average movement metric aligned –4 to +4 s from a behavioral transition. The six most frequent behavioral transitions are plotted. Lines and shaded region represent mean and SEM. (**H**) Same as **G** but with hORX-GCaMP6s signal.

## Movement tracking within different frequencies and phases of HON population signal

Recorded HON dynamics varied over multiple timescales, exhibiting sub-second to several-minute oscillations. Due to this observation, we asked if HON encoding of movement was similar across frequency domains of HON activity. Empirical mode decomposition (EMD) adaptively decomposes a signal into a set of intrinsic mode functions (IMFs) which provide a time-frequency representation of the data, and is robust to the non-stationary and non-linear nature of biological signals (*Huang et al., 1998*). Each IMF can be assigned a characteristic frequency which approximates the average oscillation in that signal component. Thus, we were able to 'break down' hORX-GCaMP6s photometry into a

set of simple IMFs, each capturing a different characteristic frequency of HON dynamics (*Figure 3A*). When plotting the average power of each HON IMF versus its characteristic frequency, we noted that maximum power generally occurred around 0.1–0.01 Hz (*Figure 3B*). A parallel EMD performed on the body movement metric revealed a maximum power in a similar frequency band (*Figure 3C*), suggesting that body movement and HONs have similar spectral profiles.

Do movements occur at specific phases within the HON activity rhythms? To examine this, we performed a phase preference analysis which tells us when, on average, a movement event occurs in the cycle of a HON oscillation. By convention, within each HON IMF, we defined the crest ('upstate') to

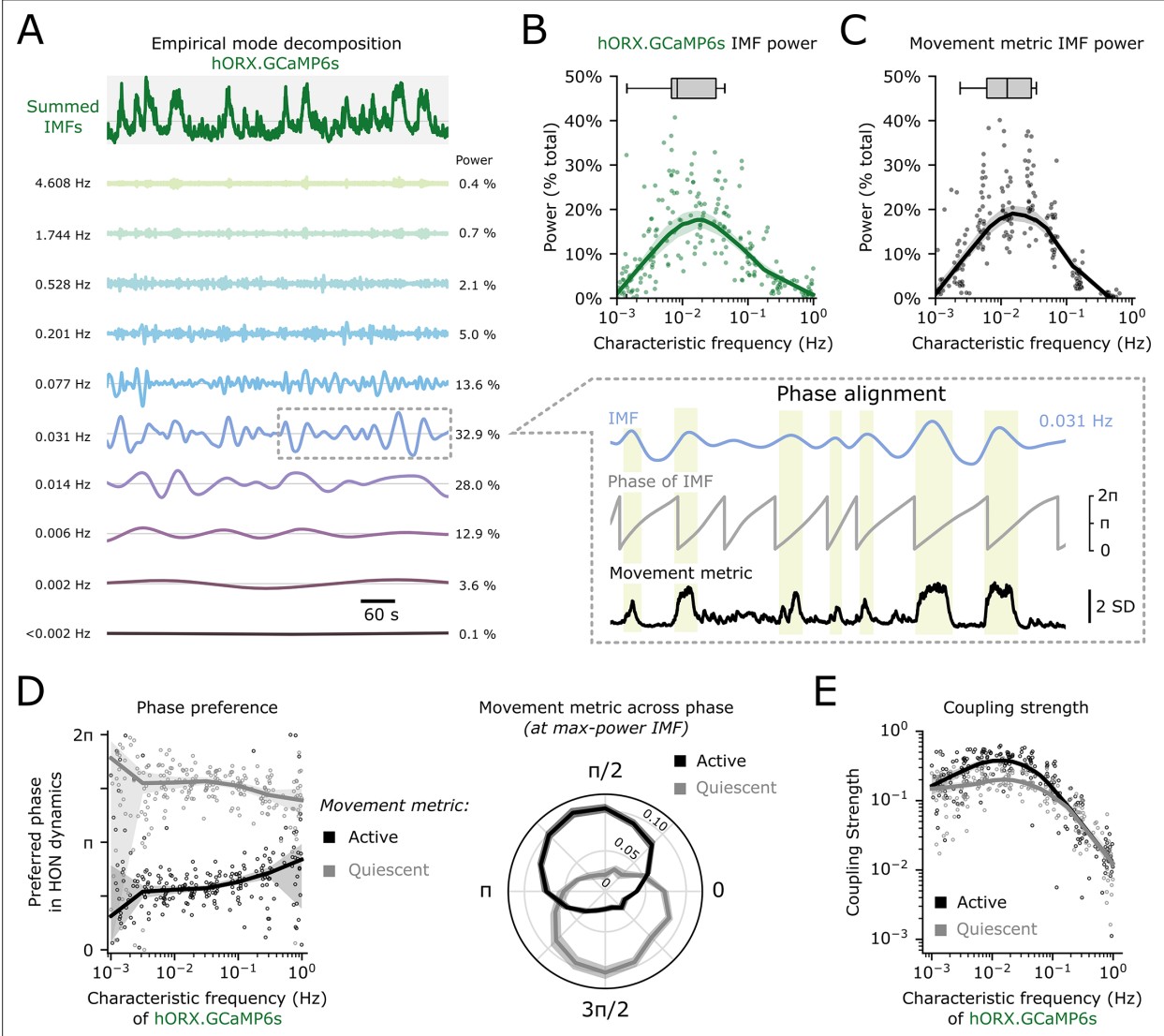

**Figure 3.** Movement magnitude is phase-aligned to hypocretin/orexin neuron (HON) activity across the frequency domain. (**A**) Representative empirical mode decomposition (see methods) of HON GCaMP6s activity from one experiment. The top green trace displays a summation of all intrinsic mode functions (IMFs). Below are the first 10 IMFs sorted by characteristic frequency in Hz. Relative power is reported as a percentage of the total. Dashed gray box (middle right) represents a cutout of the 0.031 Hz IMF in which movement epochs (shaded beige regions) are clearly phase-aligned in the 0-π period. (**B**) Average power plotted against characteristic frequency of n=229 GCaMP6s-derived IMFs derived from 27 experiments using 15 mice. Thick line and shaded region represent a local regression and 95% CI. Boxplots represent the maximum power IMF from each experiment. (**C**) Same as **B**, using n=214 movement metric-derived IMFs. (**D**) Left: Preferred phase of both active (black) and quiescent (gray) epochs of the movement metric plotted against characteristic frequency. Dots represent n=229 GCaMP6s-derived IMFs derived from 27 experiments using 15 mice. Thick line and shaded region represent a circular regression and 95% CI. Right: average absolute value of the z-scored movement metric plotted on the radial axis against the phase of the maximum-power IMF. Lines represent mean ± SEM from n=15 mice. (**E**) Coupling strength of the movement metric to GCaMP6s IMFs in both active (black) and quiescent (gray) epochs plotted against the IMF's characteristic frequency. Dots represent n=229 GCaMP6s-derived IMFs derived from 27 experiments using 15 mice. Thick line and shaded region represent a local regression and 95% CI.

be at π/2 radians and the trough ('downstate') at 3π/2 radians. In turn, we split the movement metric (z-score) above and below zero to define 'active' and 'quiescent' movement epochs, respectively. We found that active epochs preferred the π/2 HON phase while quiescent epochs preferred the 3π/2 HON phase, especially in the 0.1–0.01 Hz frequency domain (*Figure 3D*). Strongest phase coupling between HONs and movement was also observed in this frequency domain (*Figure 3E*).

Finally, we sought to confirm whether HON simply tracked, or also shaped, the body movements characterized in *Figure 3C*. We reasoned that if HONs only tracked the movements, then HON ablation would not substantially alter the power spectrum of the movements; conversely, if HONs shaped the movements, then HON ablation would alter the movements. We selectively ablated HONs using an extensively validated HON-DTR mouse model (*Viskaitis et al., 2024*; *González et al., 2016b*; *Viskaitis et al., 2022*; *Figure 4A and B*). Maximum-power IMFs were found in a similar frequency range peaking around 0.1–0.01 Hz in HON-ablated and control mice (*Figure 4C*), suggesting HONs are not necessary to maintain movement frequency profiles. To further explore this using an alternative analysis, we used a K-Means clustering algorithm to sort self-initiated movements into two broad categorical clusters constituting 'small' and 'large' movements (*Figure 4D*). The frequency of 'small' and 'large' movements was not different across control and HON-ablated mice (*Figure 4E*). Overall, these results define frequencies and phases of HON activity that track body movement information.

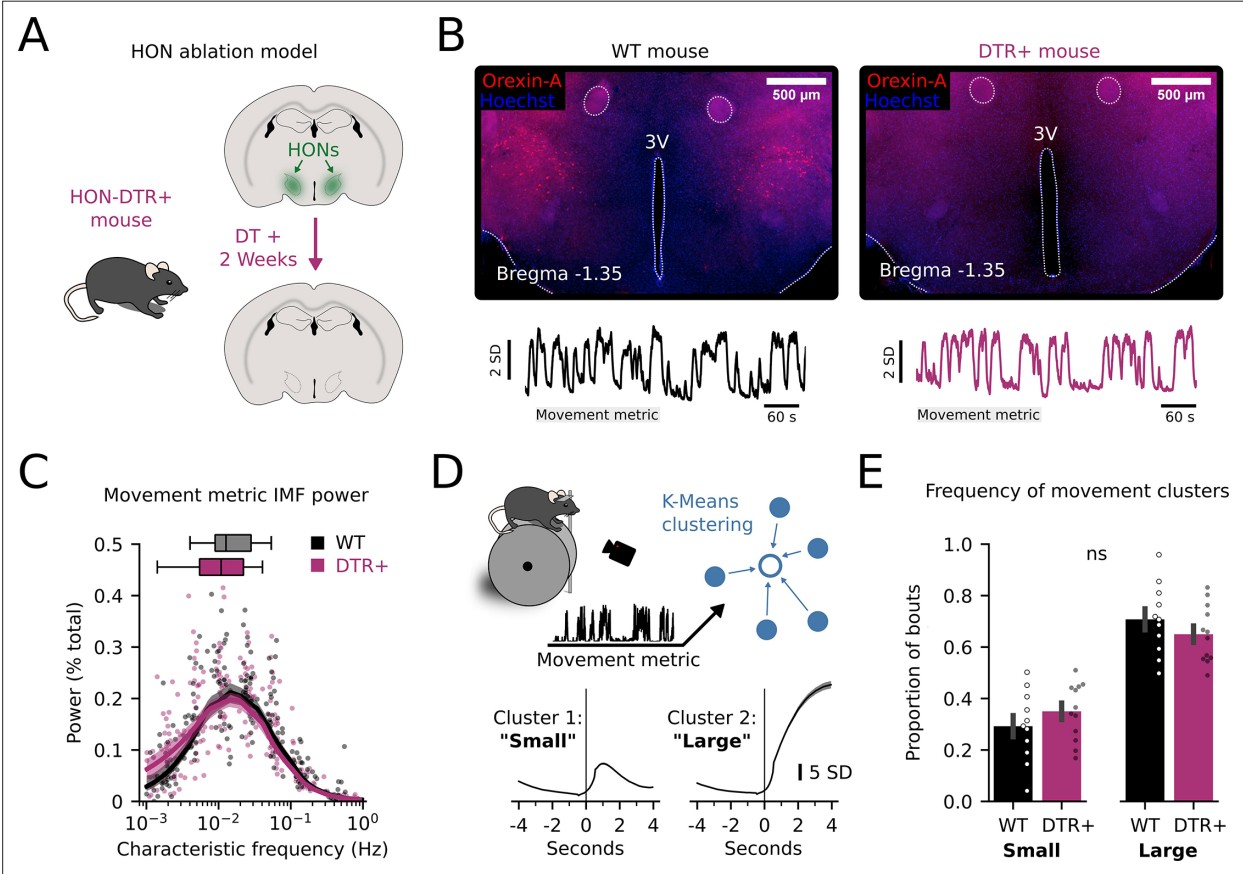

**Figure 4.** Hypocretin/orexin neurons (HONs) are not required to maintain movement profiles. (**A**) HON ablation in HON-DTR+ mice. (**B**) Brain slice histology of a wild-type mouse (Wild-type WT, left, image representative of n=11 mice) and a HON-ablated mouse (DTR+, right, image representative of n=12 mice) stained for OrexinA (red). 3V=3rd ventricle. (**C**) Average power plotted against characteristic frequency of movement metric-derived intrinsic mode functions (IMFs) (WT: n=217 IMFs from 30 experiments using 11 mice. DTR + n = 267 IMFs from 34 experiments using 12 mice). (**D**) Diagram of K-Means clustering to generate two clusters of movement representing voluntary initiation motion following >4 s of no motion, see methods. (**E**) Proportion of bouts which were assigned 'small' or 'large' were not significantly different across genotypes (unpaired *t*-test, *t*=1.1167, p=0.2767).

# Orexinergic representation of movement at different HON projection targets

Monoaminergic nuclei producing dopamine and noradrenaline, substantia nigra pars compacta (SNc) and locus coeruleus (LC), respectively, have long been studied for their involvement in arousal and movement (*Aston-Jones and Waterhouse, 2016*; *Bear et al., 2015*). Both are densely innervated by HON axons and express orexin receptors (*Peyron et al., 1998*; *Marcus et al., 2001*). Do HONs transmit movement information equally to these projection targets? To investigate this, we expressed the genetically-encoded orexin peptide biosensor OxLight1 (*Duffet et al., 2022*) into the SNc and LC for simultaneous dual-site recording in the same mouse (*Figure 5A–C*). We found that the 'active' epochs of the movement metric differed in phase preference to OxLight1 signal between the brain regions: SNc orexin recordings showed a phase preference to the 'downstate' ($3\pi/2$) phase, while LC orexin recordings preferred the 'upstate' ($\pi/2$) phase, particularly in the 0.1–0.01 Hz frequency band (*Figure 5D*), with the highest phase coupling in the lower frequency bands (*Figure 5E*). This suggests that orexin release may heterogeneously represent body movement at different HON projection targets. When orexin dynamics were aligned to 'small' and 'large' movement clusters, we observed rising signals in the LC, but falling signals in the SNc, which was especially apparent in association with 'large' clustered movements (*Figure 5F*). Together, these analyses indicate that orexin peptide release related to movement is non-homogeneous across two major HON projection targets.

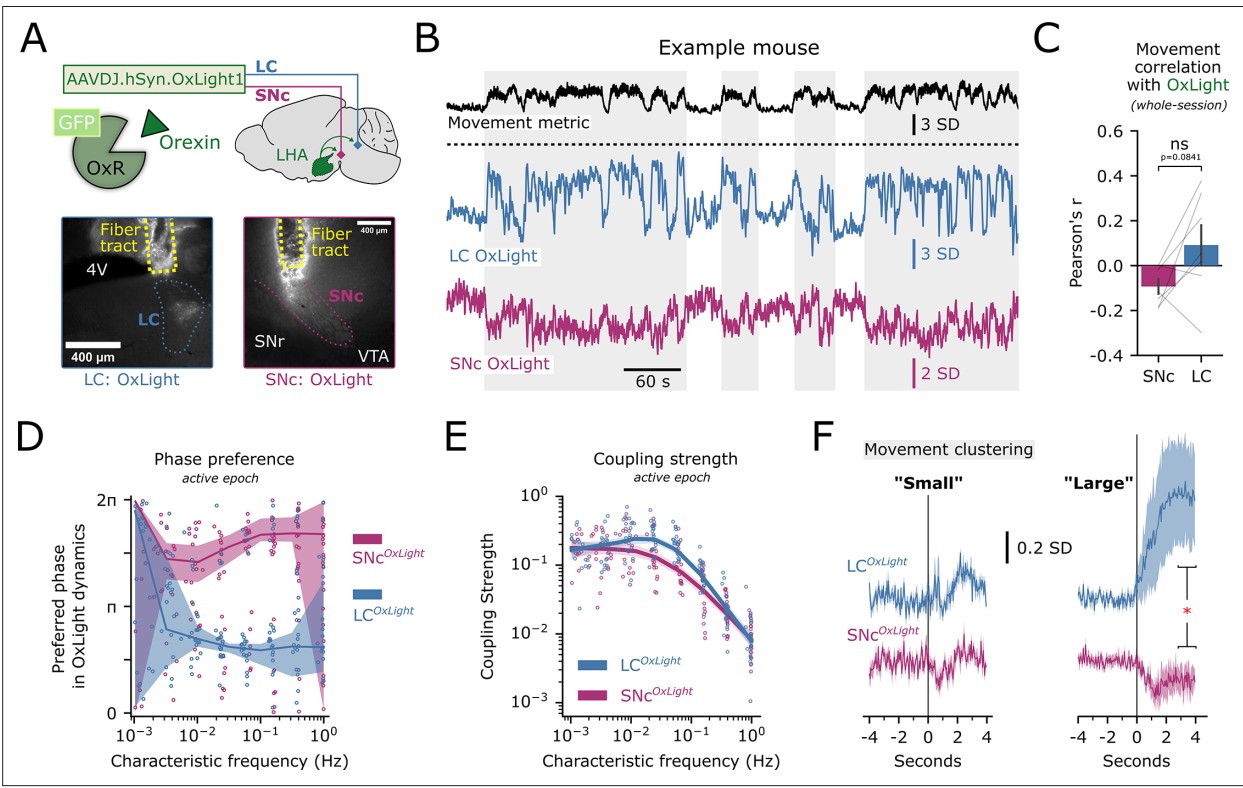

**Figure 5.** Movement-related orexin peptide release is non-homogenous across projection targets. (**A**) Schematic representing targeting of the orexin sensor OxLight1 to the locus coeruleus (LC) and substantia nigra pars compacta (SNc), representative histology from n=7 mice. (**B**) Example experiment displaying the movement metric and OxLight1 signal in the LC (blue) and SNc (purple) in the same mouse. Shaded regions represent movement epochs. (**C**) Mouse-averaged whole-session correlation of movement with OxLight1 photometry in the SNc and LC (paired *t*-test: $t_6$=2.068, p=0.0841). (**D**) Preferred phase of individual empirical mode decomposition (EMD)-derived OxLight1 intrinsic mode functions (IMFs) to active epochs of the movement metric plotted as a function of the IMF's characteristic frequency. n=317 IMFs from 21 experiments using 7 mice. Thick line and shaded region represent a circular regression and 95% CI. (**E**) Coupling strength of individual EMD-derived OxLight1 IMFs to active epochs of the movement metric plotted as a function of the IMF's characteristic frequency. n=317 IMFs from 21 experiments using 7 mice. Thick line and shaded region represent a local regression and 95% CI. (**F**) Photometry baselined –3 to –1 s before movement initiation in two clusters: small movements (left) and large movements (right). Lines and shaded regions represent mean ± SEM of n=6 mice. Asterisk represents a paired statistical comparison of mean signal 3–4 s after movement initiation (Wilcoxon test, $D_6$=1.0, *p=0.03125).

## HON movement tracking across metabolic states

Given the robust correlation of HON activity and movement across the behavioral and frequency spectra, we sought to contrast our findings against another major physiological factor thought to influence HON activity, the metabolic state (*Sakurai et al., 1998*; *Yamanaka et al., 2003*; *Burdakov et al., 2005*). First, we tested whether movement encoding by HONs depended on fasting state, which is known to strongly modulate HONs (*Sakurai et al., 1998*; *Cai et al., 1999*). We found that correlations of HON activity and movement were not significantly different in ad-libitum fed, 18h-fasted then re-fed, and 18 hr-fasted mice (*Figure 6A*). Second, we sought to quantify how the HON movement encoding compares to the recently described HON encoding of lower-frequency blood glucose information, namely the negative relations between HONs and first temporal derivative of blood glucose (*Viskaitis et al., 2024*). Using a small number of mice from this pre-existing dataset *Viskaitis et al., 2024* in which video recordings were also available (*Figure 6B*), we again performed an EMD to align individual HON IMFs to both movement and the first derivative of blood glucose after intragastric infusion of glucose (0.24 g/mL, 900 µL over 10 min; *Figure 6C*). To explore and formally define the HON signal frequency bands that carry information about movement and blood glucose, we fit each HON IMF using a multivariate model with glucose derivative and movement as weighted predictors of each IMF. We plotted the two fitted weights against corresponding IMF frequency (*Figure 6D*). Large negative weights for the blood glucose derivative were assigned to lower frequency IMFs, while large positive weights for movement were assigned to higher frequency IMFs (*Figure 6D*). When binning across frequencies, statistical comparison of these weights indicated that movement had significantly different weights than glucose derivative both in the 0.1–0.01 Hz and <0.001 Hz frequency bands (*Figure 6E*). Notably, glucose derivative did not have significant weights in the higher frequency band, whereas movement did; conversely, movement did not have significant weights in the lower frequency band, whereas glucose derivative did (*Figure 6E*). Overall, these data indicate that movement representation in the HON population signal is spectrally orthogonal to fasting state or blood glucose dynamics, and is carried selectively in the high-frequency (0.1–0.01 Hz) band of the signal, while glucose information is contained in lower frequency (<0.001 Hz) band (*Figure 6F*). Finally, we examined if movement encoding by HONs changed as a function of the blood-glucose percentile. By examining the interaction term from linear regressions fit to the n=6 mice, we observed that movement encoding by HONs was broadly independent of blood glucose percentile (*Figure 6G*).

Outside of metabolism, it has long been known that HON activity co-varies with arousal (*Mileykovskiy et al., 2005*; *Lee et al., 2005*). We, therefore, wondered how HON movement encoding compares to their encoding of arousal. To explore this, we quantified arousal by tracking pupil size (an established metric of arousal in both mice and humans *Bradley et al., 2008*; *Grujic et al., 2024*). In these recordings, we also quantified ocular movements. We then performed multivariate analysis to disentangle and compare statistical contributions of arousal vs body and eye movements to HON activity (*Figure 7A*). In confirmation of previous results (*Grujic et al., 2023*), simple regression confirmed that pupil size was positively correlated with HON activity (*Figure 7B*, r=0.45 ± 0.08). Ocular movements had a smaller, but still positive correlation (r=0.26 ± 0.02). Using drop-one feature analysis on a partial least squares regression, we found body movement was the most important (highest % contribution) feature to predicting HON activity in our experimental setup (*Figure 7C*). Expanding this analysis across the frequency domain, an empirical mode decomposition of HON activity revealed that pupil size was maximally encoded at a lower frequency than body movement (*Figure 7D and E*). In contrast, ocular movements were encoded at a higher frequency than body movements. Overall, these results suggest that HON encoding of movement is distinct from their encoding of pupil-linked arousal.

## Movement tracking in HONs versus other genetically defined subcortical neural populations

Finally, we sought to contrast the movement encoding in the HON population against other genetically defined subcortical neural clusters. We selected dopaminergic neurons of the SNc, glutamatergic neurons of the medial vestibular nucleus (MVe), and noradrenergic neurons of the LC, due to their links to HON and previously proposed roles in movement and/or arousal (*Horowitz et al., 2005*; *Carter et al., 2010*; *Korotkova et al., 2003*; *da Silva et al., 2018*; *Hagan et al., 1999*; *Horvath et al., 1999*). Additionally, we examined SF-1 expressing neurons in the ventral medial hypothalamus

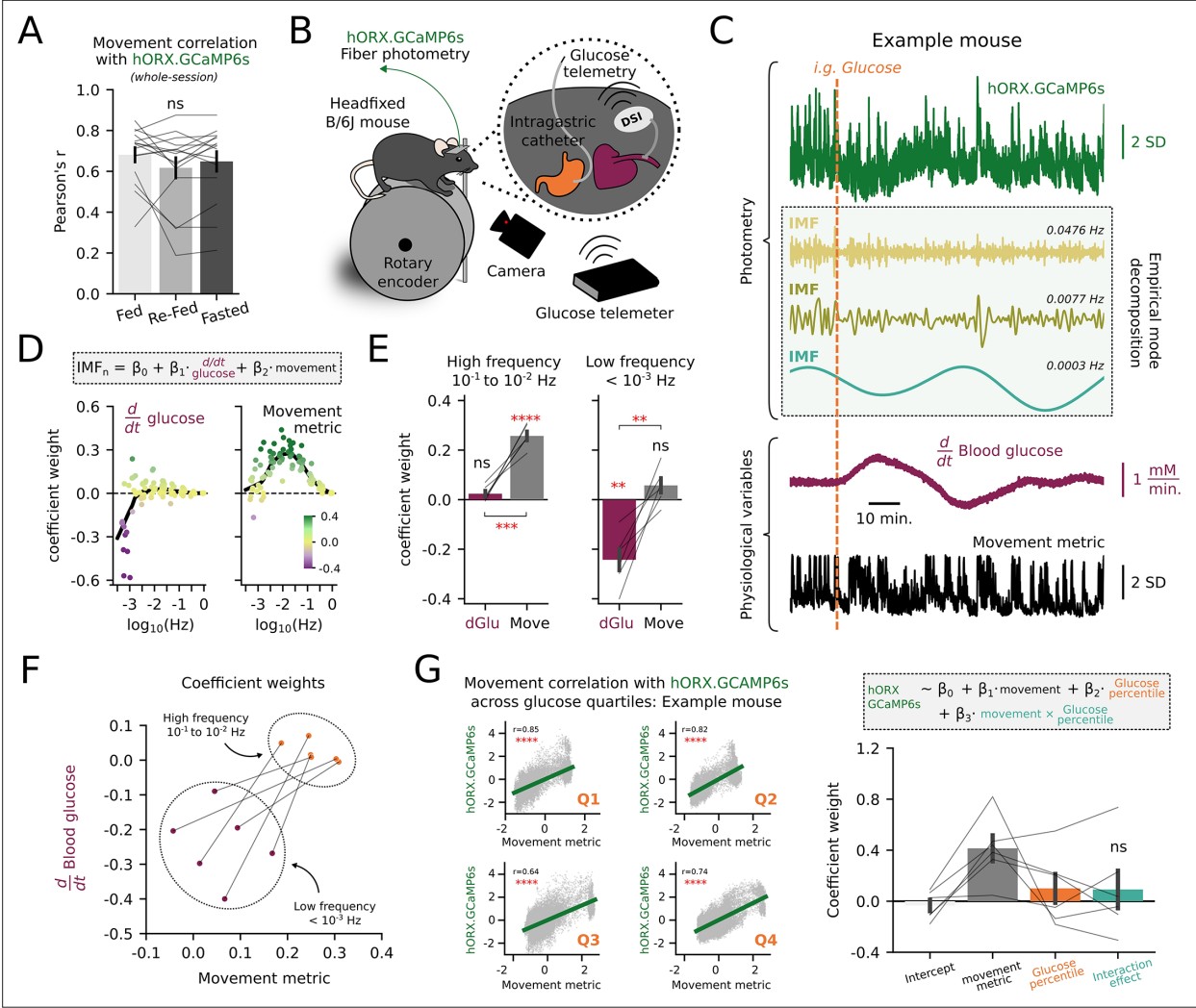

**Figure 6.** Hypocretin/orexin neurons (HONs) multiplex movement and blood glucose across frequency domains. (**A**) Correlation of movement with HON photometry across three states of fasting (n=15 mice). Fed: ad-libitum access to chow. Re-fed: 18 hr fasted and then food returned 2 hr before experiment. Fasted: 18 hr fasted mice (rmANOVA: $F_{2,28}$=1.848, $P$=0.176). (**B**) Scheme of experimental setup including photometry, glucose telemetry (DSI), video capture, and intragastric catheter infusions (*Viskaitis et al., 2024*). (**C**) Example HON GCaMP6s photometry trace (top), three derived intrinsic mode functions (IMFs) (middle), and simultaneously recorded physiological variables of blood glucose (plotted as the first derivative) and movement magnitude (lower). A vertical dashed orange line represents an intragastric infusion of glucose. (**D**) A linear model was fit to predict n=94 IMFs using the glucose derivative and movement. Fitted coefficient weights are plotted as a function of the IMF's characteristic frequency. (**E**) Per-mouse (n=6) binned coefficient weights. For the higher frequency bin (left, $10^{-1}$ to $10^{-2}$ Hz), coefficient weights were tested against a population mean of zero (Bonferroni-corrected one-sample $t$-test: glucose derivative $t_5$=1.965, p=0.2131; movement $t_5$=14.102 p<0.0001). Coefficient weights of glucose derivatives were compared to movement (paired $t$-test: $t_5$=8.331, p=0.0004). For the lower frequency bin (right, <$10^{-3}$ Hz), coefficient weights were similarly tested against a population mean of zero (Bonferroni-corrected one-sample $t$-test: glucose derivative $t_5$=−5.631, p=0.0049; movement $t_5$=1.963, p=0.2139). Coefficient weights of glucose derivative were compared to movement (paired t-test: $t_5$=5.385, p=0.0030). Bar plots display mean ± SEM. (**F**) Alternate visualization of **E** with coefficient weights plotted against each other. (**G**) Left: correlations of HON GCaMP6s signal and the movement metric across blood-glucose quartiles in an example mouse. Right: Regression predicting HON GCaMP6s activity using movement and blood-glucose percentile, performed per-mouse (interaction effect; one-sample $t$-test: $t_5$=0.6250, p=0.5594). Bar plot displays mean ± SEM. *p<0.05, **p<0.01, ***p<0.001, ****p<0.0001 and ns, not significant by two-tailed tests.

(VMH), which are thought to be sensitive to metabolic state (*Choi et al., 2013*; *Viskaitis et al., 2017*; *Tong et al., 2007*; *Meek et al., 2016*; *Deem et al., 2023*). We used different BL/6J-derived mouse cre-lines to express cre-dependent GCaMP6s activity indicators in each of these neural clusters, obtaining simultaneous fiber photometry neural activity and body movement recording in each mouse group, in the same way as for HONs (*Figure 1A*). We found that, like HONs (LHA$^{ORX}$, r=0.68 ± 0.04), the MVe glutamatergic neurons (MVe$^{GLUT}$, r=0.70 ± 0.06), correlated with movement across the entire

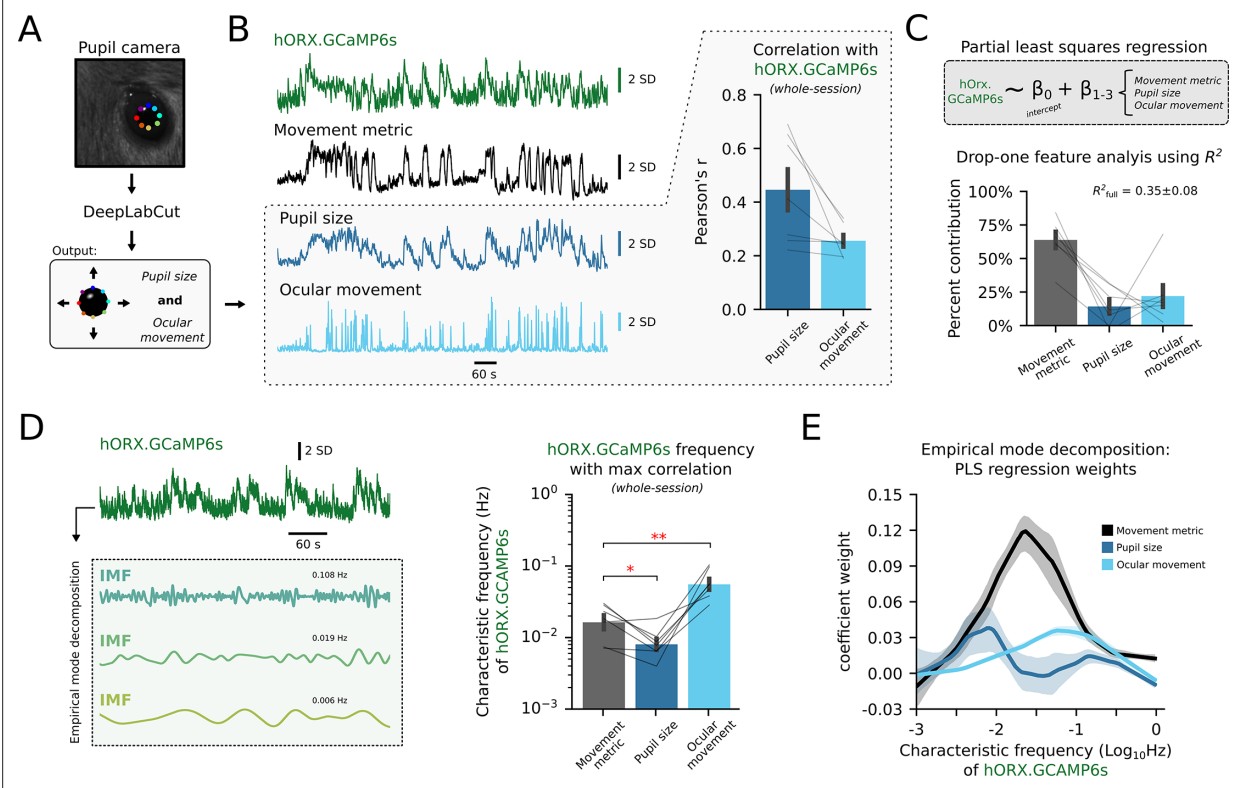

**Figure 7.** Disentangling contributions of arousal vs movement to hypocretin/orexin neuron (HON) activity. (**A**) Scheme of pupillometry processing pipeline. (**B**) Example aligned recording of HON GCaMP6s photometry (green), movement metric (black), pupil size (dark blue), and ocular movement (light blue). Right inset shows correlation of pupil size and ocular movement with HON photometry (n=7 mice). (**C**) Partial least squares regression fit to predict HON GCaMP6s photometry from the movement metric, pupil size, and ocular movement. Percent contribution of each feature is quantified using drop-one feature analysis (see methods). (**D**) Empirical mode decomposition of HON GCaMP6s intrinsic mode functions (IMFs) to quantify the frequency of HON activity containing the maximum correlation with each feature. Movement metric maximally correlated with HON frequencies that were higher than that of pupil size, but lower than that of ocular movements (Bonferroni-corrected paired $t$-test: against pupil size $t_6$=3.4199, *p=0.0283; against ocular movements $t_6$=4.6059, **p=0.0073). (**E**) Local regression displaying coefficient weights from a partial least squares regression fit to individual HON GCaMP6s IMFs, plotted as a function of the IMF's characteristic frequency.

recording (*Figure 8C*). The strength of the correlation was not significantly different across these two populations ($P$=0.8366). In contrast, LC$^{NA}$ had only a weak positive correlation (r=0.23 ± 0.07), while SNc$^{DA}$ had a weak negative correlation (r=–0.28 ± 0.08). VMH$^{SF-1}$ neurons had only a slight positive trend (r=0.20 ± 0.11). Cross-correlation between GCaMP6s and the movement metric showed that HON activity on average preceded movement by a few samples (–0.34±0.12 s), as did the LC$^{NA}$ and SNc$^{DA}$ (–0.72±0.15 and –1.08±0.52 s, respectively). MVe$^{GLUT}$ activity was closely aligned to movement (–0.15±0.09), and VMH$^{SF-1}$ activity followed movement by several seconds (4.23±0.15) using the 1.5× interquartile range to filter for outliers (*Figure 8D*). We then repeated the clustering analysis described in *Figure 4*, aligning the photometry signals to initiation of movements assigned to 'small' or 'large' movements (*Figure 8E*). For large movements, HON activity encoded movement similarly to MVe$^{GLUT}$ neurons (p=0.3915, *Figure 8F*). However, for small movements, HON activity encoded the movement metric significantly better than MVe$^{GLUT}$ neurons (p=0.0099, *Figure 8G*). Together, these data suggest HONs are uniquely positioned to track body movement to a degree not observed in the other subcortical populations we examined.

## Discussion

We found that, across behaviors, HONs tracked the amount of body movement with a high degree of precision. This tracking occurred in a higher frequency bandwidth than their tracking of blood glucose, contributing to effective multiplexing of these two fundamental physiological variables. The

HON movement tracking was not significantly different across hunger and glycemic states, suggesting orthogonal multiplexing of the movement and energy states. At two key projection targets of HONs, orexin/hypocretin peptide output was also related to movements, but in a projection-specific way, implying a heterogeneity in the movement-related HON outputs. Among the multiple neurochemically-defined subcortical neural clusters examined here, HONs were dominant in the precision of their movement tracking.

What would be the evolutionary advantages of coupling body movement to proportional changes in HONs? A central hypothesis for the meaning of HON activation is increased arousal, with evidence including electroencephalography (*Adamantidis et al., 2007*), pupil dilation (*Grujic et al., 2024*; *Grujic et al., 2023*), and cardiorespiratory adaptations (*Chen et al., 2000*; *Shahid et al., 2012*; *Williams and Burdakov, 2008*; *Kuwaki, 2008*). Though our projection-specific data imply that there

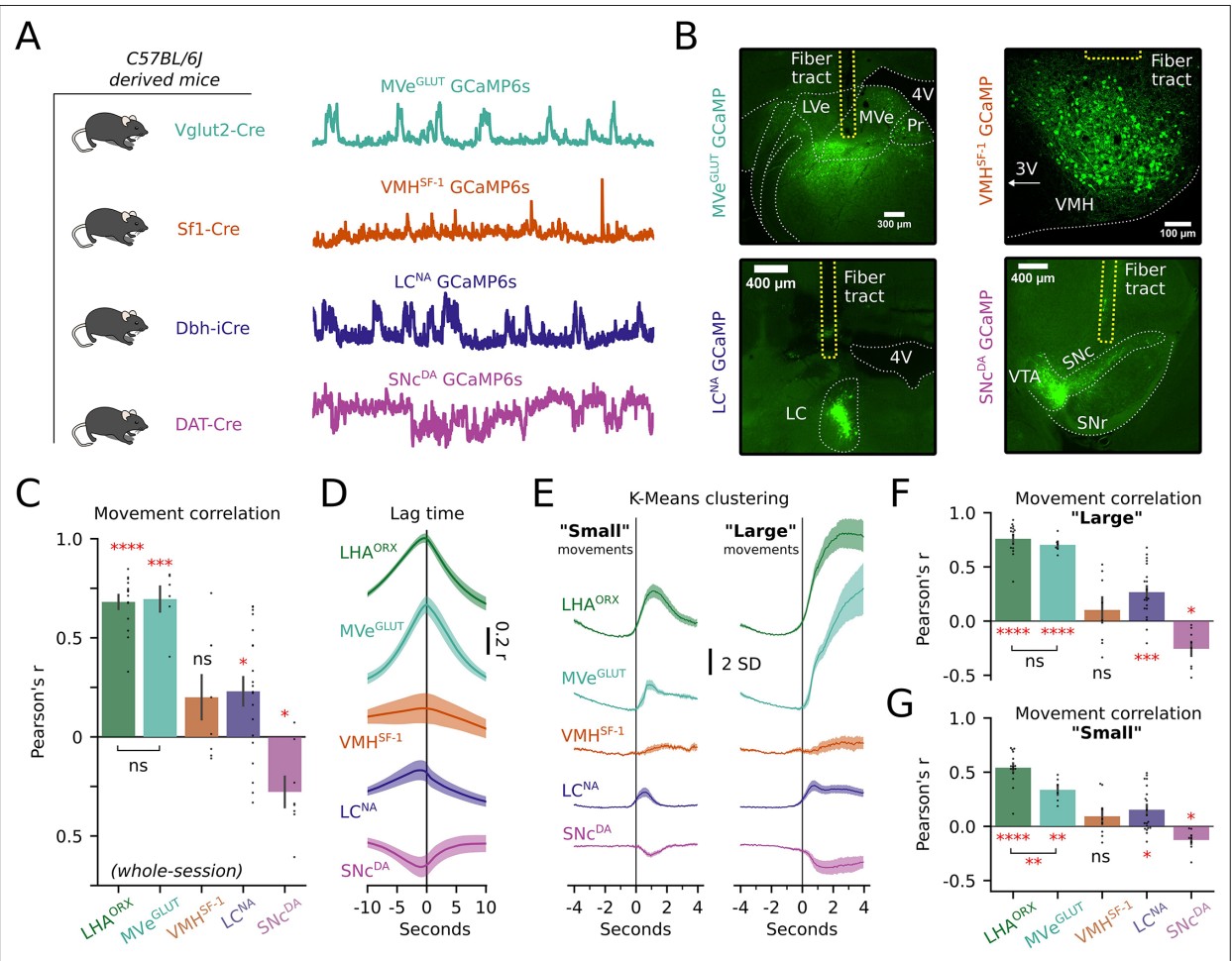

**Figure 8.** Representation of movement across genetically defined neurons. (**A**) C57BL/6J-derived strains of mice permitted Cre-dependent photometry recordings from four separate neural populations (MVe^GLUT, n=6; VMH^SF-1, n=8; LC^NA, n=20; and SNc^DA, n=8 mice). (**B**) Representative histology from each recording site. LVe; lateral vestibular nucleus, MVe; medial vestibular nucleus; Pr; nucleus prepositus; 4V; fourth ventricle; VMH; ventral medial hypothalamus 3V; third ventricle; LC; locus coeruleus; VTA; ventral tegmental area; SNc; substantia nigra pars compacta; SNr; substantia nigra pars reticulata. (**C**) Whole-session correlation of movement with photometry across neural subtypes. Asterisks represent one sample *t*-tests against mean of zero, after Bonferroni correction. Additional comparison is shown between LHA^ORX (HONs, n=15 mice) and MVe^GLUT neurons (unpaired *t*-test: $t_{19}$=0.209, p=0.8366). (**D**) Cross-correlation displaying lag time of each recorded population. Negative values imply photometry precedes movement. Lines and shaded regions represent mean ± SEM. (**E**) Photometry baselined –3 to –1 s before movement initiation in two clusters: small movements (left) and large movements (right). Lines and shaded regions represent mean ± SEM. (**F**) Correlation of movement and photometry during initiation of large movements. Asterisks represent one sample *t*-tests against mean of zero, after Bonferroni correction. Additional comparison is shown between HONs (LHA^ORX) and MVe^GLUT (unpaired *t*-test: $t_{19}$=0.877, p=0.3915). (**G**) Correlation of movement and photometry during initiation of small movements. Asterisks represent one sample *t*-tests against mean of zero, after Bonferroni correction. Additional comparison is shown between LHA^ORX (HONs) and MVe^GLUT (unpaired *t*-test: $t_{19}$=2.864, p=0.0099). *p<0.05, **p<0.01, ***p<0.001, ****p<0.0001 and ns, not significant by two-tailed tests.

may be subsets of HONs that are negatively related to movement (*Figure 5*), at the HON population level overall, the coupling between neural activity and movement was positive (r=0.68, *Figure 1*). A simple interpretation of these data is that, via the HON movement tracking, the brain creates a 'wake up' signal in proportion to movement, which could be helpful since dealing with challenges and opportunities created by movement may benefit from increased arousal. Importantly, HONs are also implicated in the mobilization of body glucose stores and shuttling glucose from liver to muscle (*Karnani and Burdakov, 2011*). Multiplexing their activation to both movement and glucose sensing may allow movement process to be optimally matched to movement energy demands.

It is tempting to speculate on the breadth at which HON population activity encodes physiological derivatives (i.e. rate of change in physiological parameters) multiplexed across the frequency domain. This study highlights how HONs represent instantaneous movement magnitude, which can be rephrased as a change in body position over time. Similarly, we previously reported population HON activity predominantly tracks the derivative of blood glucose, rather than proportionally encodes glycemic state (*Viskaitis et al., 2024*). Knowing the rate at which a physiological variable is changing allows for anticipatory responses, such as in Proportional-Integral-Derivative (PID) sensorimotor control loops, where the derivative-control component enables adaptive prediction of erroneous deviations from a set-point (*DiStefano et al., 2012*; *Aström and Hagglund, 1995*).

Do HONs 'read' or 'write' movements? From the perspective of sensorimotor control loops, the answer should be both. In theory, HONs could 'read' movements either by sensing them, or by receiving efference copies and/or corollary discharges corresponding to motor commands (*Crapse and Sommer, 2008*). Prior studies indicate that acute, targeted HON stimulation can evoke or modulate certain kinds of movement (*Karnani et al., 2020*; *Donegan et al., 2022b*; *Donegan et al., 2022a*; *Giardino et al., 2018*). Orexin receptor antagonism – or optogenetic suppression or deletion of HONs – evoked either locomotor suppression or no effect on locomotor parameters (*Samson et al., 2010*; *Li et al., 2024*; *Li et al., 2022*), consistent with our present result that the power-frequency spectrum of movements was not strongly dependent on presence of HONs themselves (*Figure 4*). Our finding that orexin level in the SNc – a site where orexin stimulates locomotion (*Viskaitis et al., 2024*) – is higher before movements rather than during movement (*Figure 5*) also suggests that at this site, orexin may be more involved in movement initiation rather than maintenance. Interestingly, in the LC, the opposite orexin profile is seen (*Figure 5*); given the role of LC in arousal, this may mean that orexin in the LC provides increased arousal during ongoing movement. Overall, in terms of movement control, HONs may be sufficient to 'write' movements, but they are not necessary. Furthermore, it is tempting to speculate on the implications of HON loss-of-function viewed from the lens of a sensorimotor control loop. Without derivative control, a system is still able to measure and perturb a variable, however, its ability to dampen unwanted oscillations from a set point is strongly impaired. From this perspective, we can speculate that narcolepsy-cataplexy, caused by HON loss of function, is perhaps explained by oscillations into unwanted sleep states and motor programs due to impaired control loops for wakefulness and movement.

In addition to these fundamental implications, there is an applied implication of our findings for basic neuroscience research. Ever since the importance of HONs for brain function and brain-body coordination was established (*Chemelli et al., 1999*; *Lin et al., 1999*; *Nishino et al., 2000*; *Hara et al., 2001*), a demand appeared for tracking their function across research and clinical settings, and many useful (but invasive) techniques for this were developed (*González et al., 2016b*; *Duffet et al., 2022*; *Mileykovskiy et al., 2005*; *Lee et al., 2005*; *Ripley et al., 2001*). The high correlation between body movement and HON activity demonstrated here suggests that video-tracked pixel difference could serve as a simple non-invasive biometric for real-time estimation of HON population state, at least as validated here in head-fixed behaving mice, a widely used model in neuroscience research (*Aguillon-Rodriguez et al., 2021*; *Guo et al., 2014*), where assessment of arousal-linked signals is in continuing demand (*Vinck et al., 2015*; *Shimaoka et al., 2018*; *Lu et al., 2020*; *Breton-Provencher and Sur, 2019*; *Hulsey et al., 2024*; *Zahler et al., 2021*).

Our results open multiple directions for further study. Upstream of HONs, it would be important to solve the challenging puzzle of how movement information is conveyed to these neurons. This challenge may likely require an extensive functional screen, because of the widespread monosynaptic inputs to HONs, with many of these coming from areas where some movement-associated activity has been reported (*González et al., 2016a*). Within the HON population activity studied here, individual

cells and their possibly heterogeneous responses to movement also remain to be defined, since the present results likely represent only a dominant subpopulation of HONs. Downstream of HONs, the findings in *Figure 5* could be extended to probe the role in movement communications of the multiple transmitters released by HONs (*Schöne et al., 2014*; *Baimel et al., 2017*; *Crocker et al., 2005*) to the genetically distinct, brain-wide targets of their terminal projections (*Peyron et al., 1998*). In future work, it will also be important to test whether movement tracking by HONs is used for optimal adjustments of the many cognitive and autonomic processes linked to HON outputs.

In summary, our findings define a key aspect of brain activity that is rapidly and precisely synchronized to even small alterations in body movement. The brain needs to sense movement for predicting future states and energy needs. Our results show that movement is encoded in parallel within glycemic state within different frequency domains of HON activity, presumably allowing these cells to communicate both energy supply and a key aspect of energy demand in the body. Deeper insights of how defined neurons and connections turn body movement into adaptive responses will enable insights into the impact of body movement on brain function in health and disease.

# Materials and methods

## Key resources table

| Reagent type (species) or resource | Designation | Source or reference | Identifiers | Additional information |
|---|---|---|---|---|
| Antibody | Anti-orexin-A (Goat Polyclonal) | Santa Cruz Biotechnology | Cat# SC-8070 RRID:AB_653610 | 1:250 dilution |
| Antibody | Anti-goat Alexa-546 (Donkey Polyclonal) | Invitrogen | Cat# A11056 RRID:AB_2534103 | 1:500 dilution |
| Strain, strain background (C57BL/6, ♂) | 'Wild type' C57BL/6J | The Jackson Laboratory | RRID:IMSR_JAX:000664 | |
| Strain, strain background (C57BL/6, ♂♀) | Vglut2-ires-Cre; $Slc17a6^{tm2(cre)Lowl}$/J | The Jackson Laboratory | RRID:IMSR_JAX:016963 | |
| Strain, strain background (C57BL/6, ♂) | SF1-Cre; Tg(*Nr5a1*-cre)7Lowl/J | The Jackson Laboratory | RRID:IMSR_JAX:012462 | |
| Strain, strain background (C57BL/6, ♂) | DAT-Cre; $Slc6a3^{tm1(cre)Xz}$/J | The Jackson Laboratory | RRID:IMSR_JAX:020080 | |
| Strain, strain background (C57BL/6, ♂♀) | DBH-iCre; Tg(Dbh-icre)1Gsc | *Stanke et al., 2006*; *Parlato et al., 2007* | MGI:4355551 | |
| Recombinant DNA reagent | AAV1-hORX-GC aMP6s.hGH | Vigen Biosciences; *González et al., 2016a* | N/A | |
| Recombinant DNA reagent | AAV9.CAG.Flex.GCa MP6s.WPRE.SV40 | Addgene | RRID:Addgene_100842 | |
| Recombinant DNA reagent | AAVDJ.hSynapsin 1.OxLight1 | UZH Viral Vector Facility | RRID:Addgene_169792 | |
| Software, algorithm | Python 3.9 | Python Software Foundation | RRID:SCR_008394 | |
| Software, algorithm | PyTorch 2.2.0 | Meta AI | RRID:SCR_018536 | |
| Software, algorithm | DeepLabCut 3.0 | https://github.com/DeepLabCut/DeepLabCut *Mathis et al., 2018*; *Nath et al., 2019* | RRID:SCR_021391 | |
| Software, algorithm | Python EMD Toolbox 0.6.2 | https://emd.readthedocs.io/en/stable/ *Quinn et al., 2021* | N/A | |

## Experimental subjects

All animal experiments followed the Swiss Federal Food Safety and Veterinary Office Welfare Ordinance (TSchV 455.1, approved by the Zürich Cantonal Veterinary Office). Adult C57BL/6J-derived lines were studied. We used previously validated cre-dependent transgenic mice: Vglut2-ires-Cre ($Slc17a6^{tm2(cre)Lowl}$/J, JAX:016963), SF1-Cre (Tg(*Nr5a1*-cre)7Lowl/J, JAX:012462), DAT-Cre ($Slc6a3^{tm1(cre)Xz}$/J, JAX:020080), or DBH-iCre (C57BL/6-Tg(*Dbh*-icre)1Gsc, MGI:4355551, *Parlato et al., 2007*). For HON ablation experiments, we used the previously validated orexin-DTR mice, wherein all HONs

were ablated via injection with diphtheria toxin >2 weeks before experiments began (*González et al., 2016b*; *Viskaitis et al., 2022*). All mice were male except for a small number of Vglut2-ires-Cre and DBH-iCre mice, which were similar to males and, therefore, pooled. Thus, whether all conclusions apply to female mice remains to be determined. Animals were housed in a 12:12 hr light-dark cycle and had ad libitum access to water and food (3430 Maintenance Standard diet, Kliba-Nafag) unless stated otherwise. All experiments were performed in the dark phase. Studies were repeated in at least two independent cohorts and, when relevant, used a semi-randomized crossover design for fasting state.

## Surgeries and viral vectors

To record photometry across a range of brain regions, we stereotaxically injected viral vectors into adult mice as follows. In all cases, mice were anesthetized with 2–5% isoflurane following pre-operative analgesia of buprenorphine and site-specific lidocaine. Viruses were stereotaxically injected at 1–2 nL/s (NanoInject III injector) either unilaterally or bilaterally at the following coordinates relative to bregma: anteroposterior (AP), mediolateral (ML), and dorsoventral (DV).

The previously validated promoter-driven AAV1-hORX-GCaMP6s.hGH (*González et al., 2016b*) ($2.0×10^{13}$ GC/mL, Vigene Biosciences) was used to record from HONs in the **LHA:** AP –1.35, ML ±0.95, DV –5.70,–5.40, and –5.10 mm. 70 nL per site. For cre-dependent mouse lines, we injected AAV9. CAG.Flex.GCaMP6s.WPRE.SV40, a gift from Douglas Kim & GENIE Project (Addgene plasmid # 100842; $1.9×10^{13}$ GC/mL, 1:5 dilution in sterile PBS) in the **MVe:** AP –6.05, ML ±0.95, DV –4.40 mm. 200 nL per site. In the **VMH:** AP –1.35, ML ±0.45, DV –5.75, –5.50, and –5.25 mm. 70 nL per site. In the **SNc:** AP –3.20, ML ±1.40, DV –4.20. 200 nL per site. In the **LC:** AP –5.35, ML ±0.90, DV –3.70 and –3.50 mm. 100 nL per site. The orexin sensor, AAVDJ.hSynapsin1.OxLight1 ($7×10^{12}$ GC/mL, UZH Viral Vector Facility, 1:3 dilution in sterile PBS) was injected into both the SNc or LC at the above coordinates. 150 nL per site. Optic fiber cannulas of 200 µm diameter, 0.39 numerical aperture (NA) with a 1.25 mm ceramic ferrule (Thorlabs) were implanted 0.1 mm above the most dorsal injection site. A custom-made aluminum headplate was secured to the skull using dental cement (C&B Metabond). Mice were given postoperative analgesia and allowed to recover for at least two weeks before the first experiment.

Expression was confirmed experimentally by observing dynamics at 465 nm excitation. A small number of mice without any observable dynamics at 465 nm, or with strong artifacts in the 405 nm isosbestic, were excluded from analysis. Expression was additionally confirmed terminally, via perfusion with 4% PFA in sterile PBS, histological slicing, and then imaging with a fluorescence microscope (Nikon Eclipse Ti2). For histological confirmation of the orexin-DTR mice, HONs were labeled with goat anti-orexin A (1:250 dilution, sc-8070 Santa Cruz Biotechnology), followed by donkey anti-goat Alexa-546 (1:500 dilution, A11056 Invitrogen) and a Hoechst stain.

For intragastric infusion surgeries, as in *Figure 6*, a small number of experiments from *Viskaitis et al., 2024* were re-analyzed if video was also available; the detailed methods for these glucose experiments are given in this source publication *Viskaitis et al., 2024*.

## Fiber photometry

For fiber photometry, we used a custom-built camera-based photometry system. GCaMP6s (or OxLight1) activity was recorded at 10–20 Hz via alternating excitation between 405 nm and 465 nm LEDs (Doric; average power 20 µW at fiber tip) in multiple animals simultaneously. The 405 nm excitation was used as an isosbestic control for movement artifacts. Most recordings were either 20 or 40 min long, except in *Figure 6*, when recording lasted for >1 hr. The first minute of each photometry trace was removed due to bleaching artifacts. Further, 465 mm fluorescence bleaching was accounted for by subtracting a triple exponential curve fit to the trace. Finally, each trace was z-score normalized to its standard deviation and mean. For visual clarity only, photometry traces in *Figures 1D and 5A*, *Figures 7B , and 8A* were smoothed using a 1 s moving average filter; all statistical comparisons and derived metrics were calculated using raw data.

## Movement metrics

All experiments were performed in a closed container illuminated via infrared LEDs to achieve constant luminance during the recording. The movement metric was generated from videos captured

using a commercial infrared camera (Blackfly S USB3, Teledyne FLIR) at the same sampling rate as photometry (10–20 Hz). To generate the metric, per-pixel differences were calculated across consecutive frames with 8-bit resolution. Then, the absolute value of the difference across every pixel was summed for each consecutive frame to generate one value for each sample, excluding the first frame. To compare this movement metric to concurrently measured photometry, the resulting trace was convolved with a 'GCaMP6s kernel' which consisted of a 60 s exponential decay kernel with a decay rate equivalent to the reported GCaMP6s half-life (1.796 s; *Chen et al., 2013*). Then, the first minute of convolution was removed to match the photometry trace and the resulting trace transformed to a z-score using its standard deviation and mean. To calculate the correlation between the movement metric and photometry, we aligned all samples from both outputs across the entire recording period, then calculated Pearson's r (scipy.stats.pearsonr, SciPy v1.14.0). Pearson's r results were averaged per mouse across repeated recording sessions. When relevant, locomotion-specific motion was recorded using an optical encoder (Honeywell, 128 ppr 300 rpm Axial) and digital acquisition box (BNC-2110, National Instruments). Locomotion was filtered using a 5-sample (250 ms) moving average filter for visualization only.

For bout-related analyses, we detected movement-bouts by first smoothing the z-scored movement-metric with a 20-sample (1 s) moving average filter and then aligning to sample indices where movement was <0 SD for at least 4 s before exceeding the threshold. For subsequent clustering, a K-Means algorithm (scikit-learn) in the default configuration was implemented to identify k = 2 clusters of movement from the identified bouts. Photometry signals were aligned to bout indices, then baselined by calculating a z-score of 1–3 s before bout initiation.

## Pupillometry

Pupil dynamics were recorded using a separate infrared camera (Blackfly S USB3, Teledyne FLIR) at a frame rate synchronized to photometry and the body-movement camera at 20 Hz. To calculate pupil metrics, we used a standard DeepLabCut (*Mathis et al., 2018*; *Nath et al., 2019*) 3.0 pipeline and manually labeled 405 video frames to identify eight points surrounding the edge of the pupil. A model was fit following 5 percent holdout validation using default recommended DeepLabCut parameters for a pretrained ResNet50 (*He et al., 2015*). Briefly, the model was trained over 200 epochs from a starting learning rate of $1\times10^{-4}$ using the Adam optimizer. The learning rate decreased by a factor of ten at epoch 160 and again at 190, following the default scheduler. After the model fit, we extracted pupil coordinates from all videos and manually inspected for accuracy. Using Python, we fit a circle (scipy.optimize.minimize, method='Nelder-Mead') to all points which DeepLabCut assigned a confidence above 0.8. During blinking, or when DeepLabCut did not return a confidence of 0.8 for at least three points, pupil size was linearly interpolated. Pupil size was defined as the area of the fitted circle. Ocular movements were defined as displacements of the fitted circle's center. Pupil size and ocular movements were then z-scored to the mean and standard deviation of the entire trace.

## Concurrent monitoring of glucose and intragastric infusions

The HD-XG telemetry system (DSI), as described in *Viskaitis et al., 2024* was used to measure and preprocess blood glucose concentrations as per manufacturer instructions. For intragastric (IG) glucose infusions, 900 µL 0.24 g/mL glucose dissolved in sterile saline was infused at a rate of 90 µL/min for 10 min. Saline controls are shown in the source publication and did not perturb HONs (*Viskaitis et al., 2024*). Missing samples were linearly interpolated, and the resulting glucose trace was resampled ( scipy.signal.resample) to align to the photometry recording Hz. To generate the derivative traces, the raw glucose trace was smoothed using a 10 min moving average filter, then resampled to align with the photometry recording Hz before the first derivative was calculated.

## Empirical mode decomposition

The logic of empirical mode decomposition analysis was largely based on *Crombie et al., 2024*, and was performed using the default settings of the Python EMD toolbox version 0.6.2 (*Quinn et al., 2021*). Briefly, traces were first normalized (z-score) before IMFs were calculated using the SIFT algorithm (*Huang et al., 1998*). The instantaneous phase, amplitude, and frequency across each IMF were calculated using the Normalized Hilbert Transform Method. The characteristic frequency was determined by taking the mean of each IMF's frequency data weighted per-sample by the IMF's amplitude

data. Extremely low-frequency ($<10^{-3}$ Hz) IMFs were excluded as they did not complete enough cycles to allow sufficient sampling for the following statistics. Power was taken as the means of the squared amplitude data and expressed as a percentage of the total power from each IMF. The preferred phase (direction) and coupling strength (length) for each IMF's phase data was then calculated from the first trigonometric circular moment with weights corresponding to positive ('active') or negative ('quiescent') signed samples of the movement metric. Note that in the 'active' case, for example, all negative values of the movement metric would be assigned weights of zero.

## Behavioral classification

To classify behaviors, we used the video captures and manually labeled five behavioral classes: chewing, grooming, resting, running, and sniffing, from grayscale frames using the previous, current, and subsequent frames to form an RBG input image. 4221 manually labeled frames were split into 'train' and 'test' datasets, stratified across behavioral classes, following ten-percent holdout validation. We augmented the dataset using random horizontal and vertical flips of the input frames to ensure robustness of the behavioral classification. A convolutional neural network was constructed using PyTorch 2.2.0: we used the standard ResNet18 with pretrained weights on the ImageNet dataset as previously described (*He et al., 2015*), but with a final linear layer corresponding to the five output classes. To train the model, we used an Adam optimizer with an initial learning rate of $1\times10^{-4}$, and a cross-entropy loss function. A scheduler (torch.optim.lr_scheduler.StepLR) decreased the learning rate by a factor of 10 every 10 epochs. The model was fit on 41 total epochs using a Tesla T4 GPU (NVIDIA) using a batch size of 128. The model was independently fit several times and consistently achieved over 98.6% accuracy on the 'test' dataset, with a cross entropy loss of <0.054. The trained network was then used to classify photometry-synchronized videos. The model output was passed through a softmax function, and thereby consisted of n video frames × 5 behavioral likelihoods. We smoothed the model-predicted likelihoods for each behavior using a 4 s moving average filter and took the max-likelihood behavior at each frame as the assigned classification. Finally, the videos and predictions were manually inspected for accuracy.

## Data analysis and statistics

Raw data were processed with custom scripts in Python 3.9 (see https://osf.io/6d45m), using Seaborn and Matplotlib libraries for plot generation. Statistical analysis was similarly performed in Python using Pinguoin v0.5.4, SciPy v1.14.0, and scikit-learn 1.2.1 libraries. In *Figures 2 and 3*, local regressions were performed using statsmodels v0.14.1 with a 0.3–0.5 fraction of the data. Confidence intervals were generated using 1000 bootstrapped samples. In the case of strictly positive variables (% power), we used a logarithm link function and transformed the result back into its original units for visualization. Circular regressions were similarly performed using scikit-learn and a multi-output support vector regression, whereby the phase (θ) in radians was evaluated over seven evenly spaced samples on the log-scale frequency by converting the phase into a multidimensional vector y:

$$y = \left[ \sin\left(\theta\right), \; \cos\left(\theta\right) \right]$$

Linear mixed effects models in *Figure 2E* were used to predict z-scored GCaMP6s photometry using the z-scored movement metric or behavioral classification. Models were defined as follows:

$$Model\,1,\; GCaMP6s_{ij} = \beta_0 + \beta_1 \times C\left(Behavior\right)_{ij} + b_{0i} + \epsilon_{ij}$$

$$Model\,2,\; GCaMP6s_{ij} = \beta_0 + \beta_1 \times Movement_{ij} + b_{1i} \times Movement_{ij} + \epsilon_{ij}$$

$$Model\,3,\; GCaMP6s_{ij} = \beta_0 + \beta_1 \times Movement_{ij} + b_{1i} \times Movement_{ij} + \beta_1 \times C\left(Behavior\right)_{ij} + b_{0i} + \epsilon_{ij}$$

Where $GCaMP6s_{ij}$ is the photometry sample, $C(Behavior)_{ij}$ is the behavioral classification, and $Movement_{ij}$ is the movement metric for the *j*-th observation of the *i*-th mouse. $\beta_0$ is the intercept, and $\beta_1$ and $\beta_2$ are the fixed effects. $b_{0i}$ is a random intercept of the *i*-th mouse. $b_{1i}$ is the random slope for movement of the *i*-th mouse. $\varepsilon_{ij}$ is the residual error. We assumed the residual error was normally distributed.

Linear regression models for *Figure 6* were calculated using the statsmodels library following the formula:

$$IMF_n = \beta_0 + \beta_1 X_1 + \beta_2 X_2 + \epsilon$$

Where $IMF_n$ is the $n^{th}$ IMF of the GCaMP6s signal, $\beta_0$ is the intercept term, $\beta_1$ and $\beta_2$ are the coefficients for $X_1$, blood-glucose derivative, and $X_2$: movement metric, respectively, and $\epsilon$ is the error term.

Partial least squares regressions in *Figure 7* were calculated using scikit-learn with n=2 components. To perform drop-one feature analysis (*Engelhard et al., 2019*), the movement metric, pupil size, and ocular movements were used as predictors to generate a 'full model.' The generated $R^2$-score (calculated as the average of a fivefold cross-validation) is denoted as $R^2_{full}$. The predictors were then removed from the model one at a time, and a 'partial' $R^2_{partial}$ was generated. The contribution of each predictor $k$ was, therefore, defined as:

$$\% \, Contribution_k = 100 \times \left( 1 - \frac{R^2_{partial,k}}{R^2_{full}} \right) \bigg/ \sum_{j=1}^{3} \left( 1 - \frac{R^2_{partial,j}}{R^2_{full}} \right)$$

Statistical tests, sample sizes, and their results are indicated in the figures, legends, and/or descriptions in the text. Where significance is presented, p-values are as follows: *p<0.05, **p<0.01, ***p<0.001, and ****p<0.0001, n.s. p≥0.05. We made no assumptions regarding the directionality of our effects; all statistical tests were two-tailed. When relevant, multiple comparisons adjustments were made using the Bonferroni correction as indicated in figure legends/or text. Data are presented as means and standard error of the mean (SEM) unless stated otherwise. Boxplots always represent the maximum, minimum, interquartile range, and median of the data.

## Acknowledgements

This work was funded by ETH Zürich. DB and ALT conceived the study and designed the protocol, with contributions from PV. ALT, PV, NG, DD, and EB performed the surgeries. ALT and PV performed most of the experiments. ALT designed and performed data analyses with contributions from PV and DD. TP provided and advised on the orexin sensor. DB and ALT wrote the text with inputs from DPR and NG. The authors have no competing interests to declare.

## Additional information

### Funding

| Funder | Grant reference number | Author |
|---|---|---|
| Swiss Federal Institute of Technology Zurich | | Denis Burdakov |

The funders had no role in study design, data collection and interpretation, or the decision to submit the work for publication.

### Author contributions

Alexander L Tesmer, Conceptualization, Data curation, Formal analysis, Investigation, Visualization, Methodology, Writing – original draft, Writing – review and editing; Paulius Viskaitis, Conceptualization, Supervision, Investigation, Methodology, Writing – review and editing; Dane Donegan, Conceptualization, Investigation, Methodology; Eva F Bracey, Investigation, Writing – original draft, Writing – review and editing; Nikola Grujic, Conceptualization, Investigation, Methodology, Writing – original draft, Writing – review and editing; Tommaso Patriarchi, Resources, Writing – review and editing; Daria Peleg-Raibstein, Resources, Writing – original draft, Project administration, Writing – review and editing; Denis Burdakov, Conceptualization, Supervision, Funding acquisition, Writing – original draft, Project administration, Writing – review and editing

### Author ORCIDs

Alexander L Tesmer https://orcid.org/0000-0002-7690-8117
Paulius Viskaitis https://orcid.org/0000-0002-9032-9867
Eva F Bracey https://orcid.org/0000-0001-8607-3312
Nikola Grujic https://orcid.org/0000-0001-8717-4123

Tommaso Patriarchi https://orcid.org/0000-0001-9351-3734
Daria Peleg-Raibstein https://orcid.org/0000-0001-6614-008X
Denis Burdakov https://orcid.org/0000-0002-9134-9165

### Ethics

All animal experiments followed the Swiss Federal Food Safety and Veterinary Office Welfare Ordinance (TSchV455.1, approved by the Zürich Cantonal Veterinary Office).

Reviewer #1 (Public review): https://doi.org/10.7554/eLife.103738.3.sa1
Reviewer #4 (Public review): https://doi.org/10.7554/eLife.103738.3.sa2
Reviewer #5 (Public review): https://doi.org/10.7554/eLife.103738.3.sa3
Author response https://doi.org/10.7554/eLife.103738.3.sa4

## Additional files

### Supplementary files

MDAR checklist

### Data availability

Source data has been deposited in a publicly available database (https://osf.io/6d45m/).

The following dataset was generated:

| Author(s) | Year | Dataset title | Dataset URL | Database and Identifier |
| --- | --- | --- | --- | --- |
| Tesmer AL | 2025 | Orexin population activity precisely reflects net body movement across behavioral and metabolic states | https://osf.io/6d45m/ | Open Science Framework, 6d45m |

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
