## [Editor Report · eLife Assessment]

This **important** study shows that the activity of hypothalamic hypocretin/orexin neurons (HONs) correlates with body movement over multiple behaviors. **Compelling** evidence, supported by sophisticated, cutting-edge tools and data analyses, highlights a link that appears to be unique to HONs. This work should be of interest to scientists studying peptidergic neurons, movement, energy regulation, and brain-body coordination.

---

## [Referee Report · Reviewer #1 (Public review)]

Summary:

This manuscript by Tesmer and colleagues uses fiber photometry recordings, sophisticated analysis of movement, and deep learning algorithms to provide compelling evidence that activity in hypothalamic hypocretin/orexin neurons (HONs) correlates with net body movement over multiple behaviors. By examining projection targets, the authors show that hypocretin/orexin release differs in projection targets to the locus coeruleus and substantia nigra, pars compacta. Ablation of HONs does not cause differences in the power spectra of movements. Movement tracking ability of HONs is independent of HON activity that correlates with blood glucose levels. Finally, the authors show that body movement is not encoded to the same extent in other neural populations.

Strengths:

The major strengths of the study are the combination of fiber photometry recordings, analysis of movement in head-fixed mice, and sophisticated classification of movement using deep learning algorithms. The experiments seem to be well performed, and the data are well presented, visually. The data support the main conclusions of the manuscript.

Weaknesses:

To some degree, it is already known that hypocretin/orexin neurons correlate with movement and arousal, although this manuscript studies this correlation with unprecedented sophistication and scale.

Taken together, this study is likely to be impactful to the field and our understanding of HONs across behavioral states.

---

## [Referee Report · Reviewer #4 (Public review)]

Summary:

Using head-fixed approach, the authors show a rapid impact of movement on the activity level of hypothalamic orexin/hypocretin neurons.

Strengths:

The head-fixed approach is great to isolate specific movements and their impact on neuronal activity.

Weaknesses:

Many of the weaknesses that were noted in the previous round of review have been addressed.

---

## [Referee Report · Reviewer #5 (Public review)]

Summary:

Hypothalamic hypocretin/orexin neurons are well-known to be involved in arousal, muscle tone and energy metabolism. Using a combination of fiber photometry, video-based movement assessments, and deep learning algorithms, the authors provide compelling evidence that the activity of these neurons correlates with net body movement over multiple behaviors and is independent of nutritional state. The authors also demonstrate that hypocretin/orexin release differs between two downstream projection sites, the locus coeruleus and substantia nigra, and are able to distinguish the activity in these sites that is due to inputs from these hypothalamic neurons vs. from other subcortical populations. The authors also convincingly show that the correlation between body movement and hypocretin/orexin neuron activity is much stronger compared to other subcortical regions. However, hypocretin/orexin neuron ablation does not affect the power spectra of movements, an observation that appears at odds with their overall conclusions.

Strengths:

The multidisciplinary approach using multiple state-of-the-art tools is supported by a rigorous experimental design and strong statistical analyses. The authors have been highly responsive to previous critiques. Concerns of another reviewer regarding the confound between arousal and movement have been addressed by new pupillometry data as a measure of arousal and multivariate analyses to distinguish between the contributions of arousal vs. movement to hypocretin/orexin neuron activity. The new data in Figure 2H added in response to a suggestion by Reviewer 3 particularly strengthens the paper.

Weaknesses:

Reviewer 2 mentioned that previous studies using orexin antagonists in rodents have largely found inconsistent effect of antagonizing orexin signaling on simple motor activity and points out that these studies are not referenced here. The authors respond that "orexin antagonism - or optogenetic silencing of HONs - evokes either reduced locomotion, or no effect on locomotor movements" and add references to paragraph 4 of the Discussion. Aside from the fact that 2 of the 3 references added are from the senior author, none address the fact that orexin antagonists induce sleep and that optogenetic silencing of these cells creates a condition where sleep can ensue with short latency - results that certainly affect body movement/locomotor activity.

---

## [Author Response]

The following is the authors’ response to the original reviews

**Reviewer #1 (Public review):**
Summary:This manuscript by Tesmer and colleagues uses fiber photometry recordings, sophisticated analysis of movement, and deep learning algorithms to provide compelling evidence that activity in hypothalamic hypocretin/orexin neurons (HONs) correlates with net body movement over multiple behaviors. By examining projection targets, the authors show that hypocretin/orexin release differs in projection targets to the locus coeruleus and substantia nigra, pars compacta. Ablation of HONs does not cause differences in the power spectra of movements. The movement-tracking ability of HONs is independent of HON activity that correlates with blood glucose levels. Finally, the authors show that body movement is not encoded to the same extent in other neural populations.Strengths:The major strengths of the study are the combination of fiber photometry recordings, analysis of movement in head-fixed mice, and sophisticated classification of movement using deep learning algorithms. The experiments seem to be well performed, and the data are well presented, visually. The data support the main conclusions of the manuscript.

We thank the reviewer for their supportive feedback.

Weaknesses:The weaknesses are minor, mostly consisting of writing and data visualization throughout the manuscript. To some degree, it is already known that hypocretin/orexin neurons correlate with movement and arousal, although this manuscript studies this correlation with unprecedented sophistication and scale. It is also unfortunate that most of the experiments throughout the study were only performed in male mice. Taken together, this study is likely to be impactful to the field and our understanding of HONs across behavioral states.

We agree that disentangling movement from arousal is an important aspect, and in the revised manuscript, we now include new data and analyses towards this (pupillometry to directly assess arousal, and multivariate analysis to assess contributions of arousal vs movemement to HON activity). In addition, we now implement many of the reviewer’s recommendations regarding writing, data presentation, and visual clarity (see our replies in the “recommendations for authors” section).

**Reviewer #1 (Recommendations for the authors):**
Some recommendations for the authors:(1) The first sentence of the Introduction states: "Neural activity related to body movement recently received much attention." I would rephrase or clarify this statement, as neuroscientists have been studying neural activity related to body movement for decades.

The reviewer is correct. Our intention was to highlight the resurgence of movementrelated neurosciences enabled by modern techniques such as deep learning applied to video data (e.g. DeepLabCut, etc). The passage has been updated for clarity.

(2) The Introduction also states that HONs orchestrate "consciousness and arousal." I would delete the word "consciousness," as consciousness represents a lofty, global concept that is challenging to define and quantify in humans, let alone mice.

We used the word consciousness to be consistent with current literature on the function of the mouse hypothalamus (e.g. Nat Neurosci 2016 Feb;19(2):290-8). But we agree it is not necessary here, and so we followed the advice to delete it.

(3) The authors state that HON dynamics were recorded while mice were head-fixed while on a running wheel. For clarity, it would be helpful to visualize this head-fixation in Figures 1A and 5B. It would also be helpful to clarify how certain behaviors (e.g. grooming, chewing) were performed and recorded while the mouse was head-fixed.

In the revised manuscript, updated graphics with a head-fixed mouse have now been added to relevant figures. Representative RGB frames (colors representing sequential frames) of each behaviour have been added to Figure 2A.

(4) In the legend for Figure 1A, the reference to Gonzalez et al. 2016 seems out of place (at least the reader should be informed why the text is referring to this previous study). Additionally, because the references are ordered by number instead of alphabetically, it would be more helpful to refer to a numbered reference rather than a name.

Gonzalez et al. 2016 references the source of the AAV construct used in this figure. This has been moved to the methods. Following eLife formatting guidelines, references will be alphabetized upon publication.

(5) In Figure 3F, it would be helpful to show visual validation that the HON-DTR method indeed ablates all HONs. This is depicted conceptually, but representative figures would be much more convincing.

A representative histological slice is now included for both wild type (WT) and HON-DTR mice in the new Figure 4B.

**Reviewer #2 (Public review):**
Summary:Despite several methodological strengths, the major and highly significant drawback is the confound of arousal with movement. This confound is not resolved, so the results could be explained by previously established relationships between orexin and arousal/wakefulness.

This an excellent point, and we agree. To address this directly in the revised manuscript, we now include new data and analyses towards this (pupillometry to directly assess arousal, and multivariate analysis to assess contributions of arousal vs movemement to HON activity).

Strengths:The authors show that orexin neuron activity is associated with body movement and that this information is conveyed irrespective of the fasted state. They also report differences in different orexin target brain regions for orexin release during movement. This paper contains an impressive array of cutting-edge techniques to examine a very important brain system, the orexin-hypocretin system. The authors offer an original perspective on the function of this system. The authors showed that orexin neuron activity scales to some degree with the magnitude of body movement change; this is unaffected by a fasted state and seems to be somewhat unique to orexin neurons.The investigation of other genetically defined subcortical neuron populations to determine the specificity of findings is also a strength, as is the ability to quantify movement and use deep learning to classify specific behaviors adds sophistication to analysis. The authors also show heterogeneity in orexin projections to specific target nuclei, which is interesting.The authors "speculate that narcolepsy-cataplexy, caused by HON loss-of-function, is perhaps explained by oscillations into unwanted sleep-states and motor programs due to impaired control loops for wakefulness and movement". This is quite an interesting aspect of their work and deserving of further study.

We thank the reviewer for their supportive feedback.

Weaknesses:Despite the strengths, there are several major and minor weaknesses that detract significantly from the study.My main concern with this work is the confound of arousal with movement so that correlations with one might reflect a relationship instead with the other. The orexin system is well known to play an important role in arousal, with elevated activity of orexin neurons reported for waking and high arousal. Orexin signaling has also been strongly associated with motivation, which also is associated with arousal and movement. The authors offer no compelling evidence that the relationships they describe between different movements and orexin signaling do not simply reflect the known relationship between arousal and motivation.The authors could address this concern by including classical arousal measurements, eg, cortical EEG recorded simultaneously with movements. Often, EEG arousal occurs independently of movement, so this could provide one approach to disentangling this confound. The idea that orexin signaling plays a role in arousal rather than movement is supported by their finding that orexin lesions using the orexin-DTR mouse model did not impact movements. In contrast, prior lesion and pharmacologic studies have found that decreased orexin signaling significantly decreases arousal and waking.Another way they could test their idea would be to paralyze and respirate animals so that orexin activity could be recorded without movement. Alternatively, animals could be trained to remain motionless to receive a reward. Thus, there are several ways to test the overall hypothesis of this work that have not been examined here.The authors propose that "a simple interpretation of their results is that, via HON movement tracking, the brain creates a "wake up" signal in proportion to movement". This seems to argue for the role of the orexin system in arousal and motivation rather than in movement per se.

Thank you. We agree that disentangling between arousal and movement is indeed critical. A classic approach is a multivariate analysis, wherein multiple simultaneously recorded “predictors” of HON activity – such as arousal and movement - can be directly compared. While EEG arousal is an option, another well-accepted metric for arousal is pupil diameter. Using n = 7 mice, we now simultaneously record HON activity, movement, running speed, pupil size fluctuations, and ocular movements:

We then fit a partial least squares multivariate regression (a regression type more robust to collinearity) using the movement metric, pupil size, and ocular movements as predictors of orexin neuron activity. Consistent with previous publications, we found that pupil size alone has a positive correlation with hORX.GCaMP6s (~0.45). However, using a drop-one feature analysis in multivariate regression, we found that movement had the highest % contribution to statistically explaining orexin neuron activity. Here are the new results (which we now added as Fig. 7A-B).

**Author response image 1. sa4fig1:** 

Furthermore, we also expanded this analysis to incorporate the different frequencies found in HON dynamics, using empirical mode decomposition. We found that pupil size had a maximum correlation at lower HON frequencies than the movement metric, while ocular movements were maximally correlated in higher frequencies (now added as Fig. 7D,E).

Overall, this analysis suggests that – while HONs encode both movement and arousal – arousal and movement do not always co-fluctuate at the same timescales, and their impacts on HONs can be disentangled in a number of ways. We now mention this in revised text on page 5.

There are several studies that have examined the effect of orexin antagonist treatment in rodents on locomotor and other motor activities. These studies have largely found no consistent effect of antagonizing orexin signaling, especially at the OxR1 receptor, on simple motor activity. These studies are not referenced here but should be taken into account in the authors' conclusions.

We agree. Prior studies found that orexin antagonism – or optogenetic silencing of HONs – evokes either reduced locomotion, or no effect on locomotor movements. We now added text and references to paragraph 4 of Discussion, summarising this.

Figure 3, panel F: I understand HON-DTR is a validated model but a picture of HONs ablation is necessary, including pictures of HONs outputs ablation within the SNc and LC.

A representative histological slice is now included for both wild type (WT) and HON-DTR mice in the new Figure 4B. Because HONs are only found in the hypothalamus, somatic deletion of HONs in this region will result in axonal degradation in output regions.

The discussion lacks a more extensive paragraph on the distinct signal and role of Ox>SNc and Ox-LC projections.

We now added sentences discussing potential implications of this to Discussion (middle of paragraph 4).

**Reviewer #2 (Recommendations for the authors):**
Minor weaknessesA very important movement in rodents is head orientation, especially given the limitation in ocular movement. However, this paper used a fixed head model which obviated this movement and did not attempt to analyze ocular movements.

Analysing ocular movements is something we had not considered but is very easy to check using pupillometry. In n = 7 mice, we recorded both orexin neurons, and ocular movements captured through an infrared camera under constant lighting. Ocular movements had a small positive correlation with orexin neuron photometry (r = ~0.26). See response to the public review above.

**Author response image 2. sa4fig2:** 

The "HON" abbreviation is not commonly used for orexin neurons, and I suggest replacing that with a more well-known abbreviation.

To the best of our knowledge, there is no universally agreed or best-known abbreviation for hypocretin/orexin neurons (we agree it would be nice if there was one!). “HONs” is a simple first letter abbreviation of hypocretin/orexin neurons, which acknowledges the two names for this peptide given by the original discoverers (de Lecea et al, and Sakurai et al, in 1998). Although this may not be the perfect abbreviation, we have kept it for now, also to be consistent with the large number (>10) of other published studies that recently used this abbreviation.

The graphs showing Pearson's r values do not demonstrate a very strong correlation between neural activity and movement change; they also lack validation of genetic expression/ablation in some cases. The results would more strongly support the conclusions if statistically significant correlations could be demonstrated between activity and movement.

We agree that a correlation of ~0.68 is probably not worthy of a “very strong” classification. While there is no universal ruleset for categorizing the strength of a correlation, we have toned down our language throughout the manuscript.

Comment regarding statistical testing of correlations: we are cautious to stand behind correlation significance testing for large sample sizes (~48’000 photometry & video samples in a 40-minute session). In our case, correlations were always extremely significant p<0.0001. The reason for this is that correlation p-values become “too big to fail” (see Lin et al. 2013) with inflated sample size. We therefore refrain from commenting on p-values and rather report between or within-subjects statistical tests, or tests against zero. See four example experiments below.

**Author response image 3. sa4fig3:** 

Citation: Lin, M., Lucas, H. C., Jr & Shmueli, G. Research Commentary—Too Big to Fail: Large Samples and the p-Value Problem. Information Systems Research 24, 906–917 (2013).

The rationale for looking at running speed, general movement, and specific types of nonlocomotor movements could be clarified and explained more thoroughly in the introduction. Why is it important to distinguish between locomotion (represented here with running) and all other movements? Presumably, this is because orexin is known to regulate arousal/locomotion. What evidence is there for orexin's role in other types of movements, which are being grouped together in Figure 1? This could be laid out in more detail in the Introduction. Relatedly, it is not very clear in the text whether the correlation between movement and orexin neuron activity includes movement related to running.

The main focus of our paper is on movement in general (i.e. video pixel difference, described in Results and Methods). This movement metric includes everything captured by the video, it is agnostic to the type of movement or behaviour. To connect this to some of the specific innate movements/behaviours typically studied in mouse literature (running, grooming, sniffing, etc), we also performed plots in Figure 2. We attempted to explain this better in revised section 1 of Results.

What exactly is being correlated in Figure 1C (and throughout the rest of the paper?) Is this the average signal correlated with the average movement change over the entire recording time? This could be more explicitly stated in methods/results. The correlations themselves/p-values could be shown in addition to/instead of Pearson's r values. Are the correlations themselves significant? This would strengthen the claim that orexin activity is strongly coupled to the magnitude of body movement change. As another example, in Figure 2D, there are no statistics reported on the correlation between movement metric and average neural signal. In Figure 6G, orexin neuron activity is more strongly correlated with movement than MVe glut neurons, but are either of these correlations significant? The correlation between MVe glut activity and movement overall seems similar to that of orexin neurons, and may be worth noting more explicitly.

Throughout the paper, we have recorded both neural activity (photometry) and movement at 20 Hz. This would generate, for example, 48’000 samples of photometry and movement from a 40-minute session. All the samples were used to calculate a pearson’s r between variables. To clarify this, we now added the subtext “wholesession” to relevant figures, as well as a clarification in the methods.

Individual experiment correlations for orexin neurons and MVe glut neurons were always significant p<0.0001, even after a Bonferroni multiple comparisons correction was applied to each population. See the “too big to fail” nature of correlation hypothesis testing above.

It could be made clearer at the end of Figure 2 that orexin neuron activity is tracking the magnitude of movement change (shown in Figure 2D), not that it is encoding different types of movement.

We intended for original Figure 2E to illustrate this concept, however this panel has caused a great deal of confusion to several readers and was perhaps ill conceived. We have replaced Figure 2E with a new panel more directly addressing the reviewer’s statement. We can construct three models where orexin neuron activity is predicted from the behavioral classification (sometimes called “one-hot” encoding) and/or the movement metric.

Model 1 predicts orexin neuron activity using only a categorical predictor of behavioral state. Model 2 only uses the movement metric, and model 3 allows a different movement-metric correlation within each behavioral state. We can compare these models using AIC (Akaike Information Criterion) which is a point estimate. While the most complex model 3 was the best, model 2 was much closer to model 3 than model 1. Similarly, model 2 was much better than model 1. From this we conclude that the magnitude of movement change is a more powerful predictor than behavioral state (“type of movement”). This is now Figure 2E.

It would be interesting to see the raw movement metric data as shown in Figures 1 and 2 in the DTR mice to show that ablating orexin neurons does not impair the movement profile seen in Figures 1 and 2.

The requested visualization has been added to Figure 4B.

Validation that orexin was selectively ablated in these mice would be ideal.

Histology (see response to public review) was added to a new Figure 4B.

Figure 4A - OxLight expression in SNc does not look very robust.

Please note this is a membrane-targeted indicator, the staining this produces is thus much weaker than cyctosolic indicators such as calcium indicator GCaMP.

Figure 4 - It would be beneficial to see the same correlations that were done in Figures 1 and 2 to show OxLight activity vs. movement metric. Are they correlated?

Individual traces had significant correlations with OxLight and movement, and the population averages revealed similar trends:

**Author response image 4. sa4fig4:** 

Figure 6B - Targeting of MVe neurons does not look very specific. The sample size for orexintargeted mice should be re-stated in the figure legend for clarity.

Legend has been updated to clarify n = 15 for orexin targeted mice.

Some citations didn't seem to match what was being referenced in the text. Similarly, in the legend for Figure 1C, the statistics do not match what is reported in the text. In Figure 1, the sample size is not noted in the text. When referring to running in Figure 1, is this referring to running speed? Perhaps the language could be more consistent.

These typos (due to a rounding error) in the legend and text have been corrected. Sample size has been added to the text, and we have changed Figure 1D to clarify we are referring to running speed. We moved some citations to improve clarity.

Methods - where were Cre mice obtained from?

Sources now better referenced in Methods (JAX or Parlato et al).

Figure 1, panel C: The authors compared Pearson's r-coefficient results for each animal and for each variable. However, it would be interesting to show the correlation curves for each variable. However, it would be interesting to show the correlation curves for each variable as well here. Also, there is mention of a strong correlation but it is unclear whether these correlations are significant.

See below for an example mouse.

**Author response image 5. sa4fig5:** 

Figure 3, panel F: I understand HON-DTR is a validated model but a picture orexin ablation is necessary, including pictures of orexin fibers ablation within the SNc and LC.

See our reply to the public review above.

Figure 5, Panel A: Same comment as Figure 1, panel C.

We have similarly clarified the panel and legend.

Page 4: The authors mention "Within the 1st and 4th quartile of blood glucose, movement-HON correlations were not significantly different. Please add the figures.

The requested plot has been added to Figure 6, panel G.

**Reviewer #3 (Public review):**
SummaryThe study presents an investigation into how hypothalamic orexin neurons (HONs) track body movement with high precision. Using techniques including fiber photometry, video-based movement metrics, and empirical mode decomposition (EMD), the authors demonstrate that HONs encode net body movement consistently across a range of behaviors and metabolic states. They test the ability of HONs to track body movement to that of other subcortical neural populations, from which they distinguish HONs activity from other subcortical neural populations.Strengths:The study characterizes HONs activity as key indicators of movement and arousal, and this method may have potential implications for understanding sleep disorders, energy regulation, and brain-body coordination. Overall, I think this is a very interesting story, with novel findings and implications about sensorimotor systems in animals. The manuscript is clearly written and the evidence presented is rigorous. The conclusions are well supported by experimental data with clear statistical analyses.

We thank the reviewer for their supportive feedback.

Weaknesses/suggestions:There are a couple of issues I think the authors could address to make the paper better and more complete:(1) The study primarily focuses on steady-state behaviors. It would be interesting if the authors' current dataset allows analyses of HON dynamics during transitions between behavioral states (e.g., resting to running or grooming to sniffing). This could provide additional insights into how HONs adapt to rapid changes in body movement.

This is a fantastic idea, and easy to check using our classification CNN. We identified the six most frequent behavioral transitions and plotted them in Figure 2H. HONs show rapid dynamics in activity aligned with behavioral changes.

These changes are very similar to the movement magnitude along these transitions, which is now also plotted in Figure 2G.

(2) Given the established role of HONs in arousal and wakefulness, the study could further investigate how movement-related HON dynamics interact with arousal states. For example, does HON encoding of movement differ during sleep versus wakefulness?

To further investigate how movement encoding interacts with arousal, we now include quantification and analysis of pupil-linked arousal (see new Figure 7). We agree it would be interesting to look at what happens during sleep, especially REM sleep when some HONs are thought to be active where there is no/little body movement, but this is beyond the scope of the present study.

(3) Although HON ablation experiments suggest that HONs do not shape movement frequency profiles. It would be more compelling if the authors could investigate whether HONs contribute to specific types of movements (e.g., fine motor vs. gross motor movements) or modulate movement initiation thresholds.

We performed this analysis using the k-means classifier for small/large movements. Consistent with previous results, we found no significant effect (p = 0.2767) of genotype on the frequency of identified small (fine) or large (gross) movement clusters. This plot has been added to Figure 4E.

(4) The heterogeneous movement-related orexin dynamics observed in the LC and SNc raise intriguing questions about the circuit-level mechanisms underlying these differences. Optogenetic or chemogenetic manipulation of these projections could validate the functional implications of these dynamics.

We agree. We now discuss some implications of this in revised Discussion (paragraph 4). Please note that previous work already demonstrated that orexin action in the SNc can produce locomotion (referenced in the paragraph), though we agree that further work would be valuable.

**Reviewer #3 (Recommendations for the authors):**
Additional feedback:(1) Figure 1C: the individual data points are hard to track or see. Consider using a larger marker face to help data visualization. Similar issues can be found in Figures 2C, 2E, 5E, 6C, 6F, and 6G.

Thickness of the lines and scatterplots have been increased.

(2) First Section of Results: the authors claim to use a deep-learning network to automatically classify video recordings into five distinct behaviors. However, several issues need to be addressed here:a. In Results, the corresponding sentence lacks a reference to the Methods Section.

Reference has been added to the text.

b. In Methods, the description of the CNN model is quite limited, lacking many basic, necessary components including necessary references to published papers, the model training, characterization (only an overall accuracy is not enough), as well as dataset definition, preparation, augmentation (if any), etc.

We have expanded the methods section regarding the CNN model.

(3) First Section of Results: in the second paragraph, the authors claim that "Overall, these results reveal HON population activity precisely tracks a general degree of body movement across recorded behaviors." This is not accurate. To indicate that HONs activity tracks the general degree of body movement across behavior states, they need to further show that behavioral states with similar levels of movement metrics can be differentiated via HON activities. However, as they showed in Figure 2D, some behaviors with similar values of movement metric do not seem to be easily discerned by HON activity levels.

We agree with you, and this is also what we originally intended to convey – now reworded for clarity.

(4) Technical issue: Figures 3B, 3C, 3G, using local regression to plot the solid lines makes them touch negative values, which does not make sense for "power proportion" (this quantity is always non-negative).

This is a good point. To fix this, we first log-transformed the power metric, then performed a local regression, and used the link function to transform the model predictions back to %-units for visualization. This has been noted in the methods.

(5) Figure 3G: For a better comparison, consider combining the two plots into a single plot.

The two plots have been merged as shown in Figure 4C.

(6) Figure 5E: For a better data visualization, the current pair of plots can be consolidated into one single plot where the x-axis is Move and the y-axis is dGlu. In this way, it is easier to understand and the orthogonality as claimed in the manuscript can be more apparent.

The requested plot has been added as Figure 6F.